# Zeroth-Order Fine-Tuning of LLMs with Extreme Sparsity

**Wentao Guo** [1 2]  **Jikai Long** [3]  **Yimeng Zeng** [4]  **Zirui Liu** [5]  **Xinyu Yang** [6]  **Yide Ran** [3]
**Jacob R. Gardner** [4]  **Osbert Bastani** [4]  **Christopher De Sa** [2]  **Xiaodong Yu** [3]  **Beidi Chen** [6]  **Zhaozhuo Xu** [3]

## Abstract

Zeroth-order optimization (ZO) is a memory-efficient strategy for fine-tuning Large Language Models using only forward passes. However, the application of ZO fine-tuning in memory-constrained settings such as mobile phones and laptops is still challenging since full precision forward passes are infeasible. In this study, we address this limitation by integrating sparsity and quantization into ZO fine-tuning of LLMs. Specifically, we investigate the feasibility of fine-tuning an extremely small subset of LLM parameters using ZO. This approach allows the majority of un-tuned parameters to be quantized to accommodate the constraints of limited device memory. Our findings reveal that the pre-training process can identify a set of "sensitive parameters" that can guide the ZO fine-tuning of LLMs on downstream tasks. Our results demonstrate that fine-tuning 0.1% sensitive parameters in the LLM with ZO can outperform the full ZO fine-tuning performance, while offering wall-clock time speedup. Additionally, we show that ZO fine-tuning targeting these 0.1% sensitive parameters, combined with 4 bit quantization, enables efficient ZO fine-tuning of an Llama2-7B model on a GPU device with less than 8GiB of memory and notably reduced latency.

## 1. Introduction

Large language models (LLMs) have demonstrated superior performance in general-purpose language generation (Brown et al., 2020; Radford et al., 2019; Liu et al., 2019). Despite their success, it remains necessary to fine-tune LLMs for specific tasks to achieve optimal results. However,

fine-tuning LLMs often requires much more memory compared to the inference process. Specifically, there are mainly four parts that occupy the memory during fine-tuning LLMs: **(1)** the weight parameter itself; **(2)** the optimizer state, which contains the information about the past gradient (Kingma & Ba, 2015); **(3)** the weight gradient used to update the parameters; **(4)** the activation cached to calculate the weight gradient (Liu et al., 2024b); In previous work like QLoRA (Dettmers et al., 2023), it can reduce both **(1)** and **(2)** by combining weight quantization and low-rank adaption (Hu et al., 2021), which enables fine-tuning huge LLMs under data-center level GPUs. However, on memory-constrained hardware like cell phones, the memory of caching **(3)** weight gradient and **(4)** activation required by backpropagation still cannot be overlooked. The disparity between the demand of LLM fine-tuning and hardware capacity limits the adaptability of LLMs, especially when personalizing them for edge devices.

**Exploring Zeroth-Order Optimization in LLM Fine-Tuning.** Recently, there has been a resurging interest in zeroth-order (ZO) optimization methods for LLM fine-tuning (Malladi et al., 2023a; Liu et al., 2024a; Chen et al., 2024). ZO optimization method perturbs model parameters in random directions and utilize the loss value difference to compute the gradient direction for parameter update. One advantage of ZO methods in LLM fine-tuning is that they do not require backpropagation procedures, which significantly saves the computation and memory. In this way, ZO is backpropagation-free and does not need to cache **(3)** weight gradients and **(4)** activations during fine-tuning. In practice, ZO methods have demonstrated the potential to achieve performance comparable to first-order methods in LLM fine-tuning, which opens the doors for various efficient LLM adaptation strategies.

**Efficient ZO LLM Fine-Tuning with Sparsity.** Although ZO methods remove the need for backpropagation, a significant drawback of these methods is the slow convergence rate (Zhao et al., 2024; Liu et al., 2024a). A recent approach addresses this by fine-tuning with a sparse mask (Liu et al., 2024a; Zhang et al., 2024), achieving approximately $\sim 75\%$ sparsity. Nonetheless, this sparsity level barely reduces computational overhead, as the latency during the forward pass with *even* $\sim 90\%$ sparsity is still comparable to that of

[1]Princeton University [2]Cornell University [3]Stevens Institute of Technology [4]University of Pennsylvania [5]Rice University [6]Carnegie Mellon University. Correspondence to: Wentao Guo <wg0420@princeton.edu>, Zhaozhuo Xu <zxu79@stevens.edu>.

Accepted to the Workshop on Advancing Neural Network Training at International Conference on Machine Learning (WANT@ICML 2024).

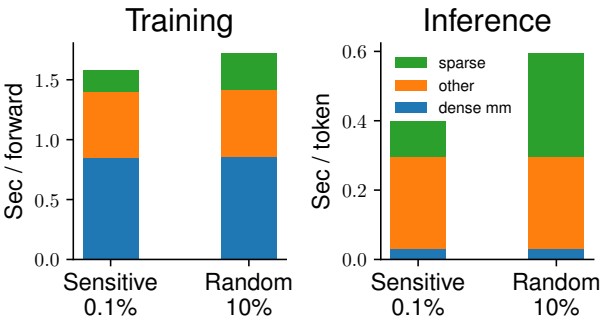

Figure 1: Training & inference speed of Llama2-7B. As the sensitive sparse fine-tuning method achieves great performance via optimizing only *0.1%* parameters (performance comparable to ZO full fine-tuning and 10% random subsets), during inference we achieve an end-to-end $1.49\times$ speedup, with $2.15\times$ speedup at sparse operations.

dense matrix operations. This latency increase can greatly impact user experience on applications such as personal assistants, where even a twofold increase in latency is perceptible. In addition, merging the sparse weights back into the base model is impractical on these devices due to memory constraints prohibiting dequantization and quantization. Empirical evidence suggests that higher sparsity levels can significantly decrease the time required for sparse matrix operations, as shown in Figure 1. This raises the question:

*Is it possible to leverage the benefits of higher sparsity levels in reducing inference latency while preserving performance on downstream tasks? If so, how far can sparsity be pushed in this context?*

**Our Proposal: ZO LLM Fine-Tuning with Fisher-Informed, Transferable Sparsity.** In this paper, we answer the raised research question by proposing an efficient sparse ZO LLM fine-tuning strategy. We observe an extreme sparsity pattern in LLM parameters: a subset, determined by selecting the top $k$ magnitude entries from the diagonal of empirical Fisher information matrix, is effective for ZO fine-tuning. Moreover, we find this sparsity pattern can be obtained through LLM's continuous pre-training process and be transferred to various downstream tasks without modification.

**Summary of Contributions.** Building on these insights, our work proposes a comprehensive framework for ZO fine-tuning, making the following contributions:

- We identify that only an extremely small portion (**0.1%**) of LLM parameters should be updated during ZO LLM fine-tuning. Moreover, we utilize this insight to guide the memory-efficient on-device personalization of LLMs by low-bit quantization of model parameters.

- We observe the sparsity pattern observed in LLM pretraining can be transferred across different downstream

tasks while still maintaining good ZO performance. Based on this observation, we develop a computational framework to perform parameter-efficient ZO fine-tuning of LLMs.

- We conduct extensive experiments across various LLMs and demonstrate that our method achieves competitive performance across various downstream tasks.

## 2. Background and Related works

In this section, we present the formulation for ZO optimization. We also discuss related works about sparsity in LLMs.

### 2.1. Zeroth-Order Optimization

**ZO surrogate gradient estimator.** ZO optimizers have been studied widely in the machine learning community. Given a dataset $\mathcal{D} = \{(\mathbf{x}_1, y_1), \ldots, (\mathbf{x}_n, y_n)\}$ and a loss function $f$ with model parameters $\mathbf{w} \in \mathbb{R}^d$, ZO optimizer will estimate the gradient at $\mathbf{w}$ via ZO surrogate gradient estimator. Simultaneous Perturbation Stochastic Approximation (SPSA) (Spall, 1992) is such an estimator that would first sample a random vector $\mathbf{z} \in \mathbb{R}^d$ and uses the *loss value difference* to scale the update direction. $\mathbf{z}$ is usually sampled from an Gaussian distribution $\mathcal{N}(\mathbf{0}, \mathbf{I}_d)$.

**Definition 2.1** (**Simultaneous Perturbation Stochastic Approximation (SPSA)** (Spall, 1992))**.** SPSA estimates the gradient w.r.t. $\mathbf{w}$ with a data example $(\mathbf{x}, y)$, a small constant $\epsilon \in \mathbb{R}$, and a sampled random vector $\mathbf{z} \in \mathbb{R}^d$ as follows:

$$\hat{g}(\mathbf{w}, (\mathbf{x}, y), \mathbf{z}) = \frac{f(\mathbf{w} + \epsilon\mathbf{z}; (\mathbf{x}, y)) - f(\mathbf{w} - \epsilon\mathbf{z}; (\mathbf{x}, y))}{2\epsilon}\mathbf{z}$$

(1)

There are other ZO surrogate gradient estimators available (Liu et al., 2020; Ohta et al., 2020), but in practice SPSA achieves good performance in ZO optimization, particularly when fine-tuning LLMs. Some ZO algorithms such as DeepZero (Chen et al., 2024) would utilize the *parameter-wise* finite difference of loss values to derive *parameter-wise* update directions. This would yield $O(d)$ query costs per training step *even when combining with certain sparse masking methods* and not practical for LLM fine-tuning scenarios. We therefore select SPSA with random Gaussian perturbation as our ZO gradient estimator.

**ZO-SGD algorithm.** ZO-SGD is an optimizer similar to SGD but replaces the FO gradient with ZO surrogate gradient estimate per training step, as defined below:

**Definition 2.2** (**ZO-SGD update rule**)**.** ZO-SGD is an optimizer that uses ZO surrogate gradient to update parameters $\mathbf{w}_t$ with learning rate $\eta_t$ and a data example $(\mathbf{x}_t, y_t)$ sampled at timestep $t$:

$$\mathbf{w}_{t+1} = \mathbf{w}_t - \eta_t \hat{g}_{\mathbf{w}}(\mathbf{w}_t, (\mathbf{x}_t, y_t), \mathbf{z}_t) \qquad (2)$$

MeZO (Malladi et al., 2023a) is a ZO-SGD algorithm that uses the "random seed trick" to save the need of caching ZO surrogate gradient. The choice of optimizer (SGD) is orthogonal to ZO optimization techniques, but in our preliminary experiments we find adaptive optimizers such as Adam (Kingma & Ba, 2015) would not necessarily accelerate ZO convergence in LLM fine-tuning scenarios. There are other ZO optimizers aware of the parameter-wise heterogeneity of loss curvatures to accelerate the optimization convergence (Zhao et al., 2024), and we leave how to combine our method with theirs as future works.

## 2.2. Sparsity in LLMs

Sparsity-driven techniques are widely adopted in improving ML model's efficiency (Tan et al., 2024a; Xia et al., 2023; Liu et al., 2023; Peng et al., 2013; Frankle & Carbin, 2019) and robustness (Zhong et al., 2024; 2021). Frankle & Carbin (2019) showed that within large feed-forward networks, there exists a subnetwork that, when trained in isolation, can achieve test accuracy comparable to that of the original network. In the foundation models era, Liu et al. (2023) demonstrated that transformer-based models, such as OPT (Zhang et al., 2022), exhibit great sparsity ($\geq 95\%$) in activations. Moreover, Panigrahi et al. (2023) discovered that for RoBERTa (Liu et al., 2019), fine-tuning a very small subset of parameters ($\sim 0.01\%$) can yield performance exceeding 95% of that achieved by full fine-tuning.

In the context of ZO optimization, Liu et al. (2024a) and Zhang et al. (2024) also suggest that sparsity would potentially accelerate ZO optimization convergence. We believe that *ZO has an intrinsic need for sparse training*, as the procedure of ZO gradient estimator usually requires *uniform coordinate-wise scale (in expectation)* perturbation which grows with $d$. In tradition, people usually resolve this with knowledge from parameter-wise loss curvature heterogeneity (replace $\mathbf{z}$ with $\Sigma^{1/2}\mathbf{z}$ where $\Sigma^{1/2}$ serves as a Hessian-informed preconditioner) (Ye et al., 2018; Zhao et al., 2024). However, they do not provide a comprehensive investigation on massive parameter models like LLMs. In particular, we also observe that during first-order (FO) fine-tuning of LLMs, *the FO gradient can be quite sparse*. We will elaborate more on this insight in the following section (see Figure 2 and Figure 7). We would like to explore how sparsity can benefit the ZO LLM fine-tuning.

# 3. Chasing Extreme Sparsity in ZO LLM Fine-Tuning

In this section, we describe the extreme sparsity pattern we observed in LLMs and how we utilize it for efficient ZO fine-tuning including on-device personalization of LLMs.

## 3.1. Extreme Sparsity Pattern in LLM

**ZO optimization with sensitive parameters.** Given model parameters $\mathbf{w}$, a loss function $f$, a data example $(\mathbf{x}, y)$, sensitive parameters are defined as *parameters whose corresponding FO coordinate-wise gradient square values are maximized*.

**Definition 3.1** (**Sensitive parameter mask**). A sensitive sparse mask $\mathbf{m}_k \in \{0, 1\}^d$ with $k$ nonzero entries ($\sum_i \mathbf{m}(i) = k$) is defined as

$$\mathbf{m}_k = \operatorname{argmax}_{\mathbf{m}} \|\mathbf{m} \odot \nabla f(\mathbf{w}; (\mathbf{x}, y))\|_2^2. \tag{3}$$

In the context of ZO optimization, we will update sensitive parameters *only*. Denote that $\bar{\mathbf{z}} = \mathbf{z} \odot \mathbf{m}_k$. We will modify the SPSA gradient estimator from $\hat{g}(\mathbf{w}, (\mathbf{x}, y), \mathbf{z})$ to $\hat{g}(\mathbf{w}, (\mathbf{x}, y), \bar{\mathbf{z}})$, and accordingly:

**Definition 3.2** (**Sensitive sparse ZO-SGD update rule**).

$$\mathbf{w}_{t+1} = \mathbf{w}_t - \eta_t \hat{g}_{\mathbf{w}}(\mathbf{w}_t, (\mathbf{x}_t, y_t), \bar{\mathbf{z}}_t) \tag{4}$$

The theoretical support of sensitive parameters can be derived from the lens of SPSA gradient estimator and Fisher information matrix as follows:

• **Maximum zeroth-order loss value changes, from the lens of SPSA estimator.**
The square (account for negativity) of loss value difference for $\hat{g}_{\mathbf{w}}(\mathbf{w}_t, (\mathbf{x}_t, y_t), \bar{\mathbf{z}}_t)$ is as follows:

$$\begin{aligned}
&\mathbb{E}_{\bar{\mathbf{z}}}\{f(\mathbf{w} + \epsilon\bar{\mathbf{z}}; (\mathbf{x}, y)) - f(\mathbf{w} - \epsilon\bar{\mathbf{z}}; (\mathbf{x}, y))\}^2 \\
&\approx \mathbb{E}_{\bar{\mathbf{z}}}\{2\epsilon\bar{\mathbf{z}}^\top \nabla_{\mathbf{w}} f(\mathbf{w}; (\mathbf{x}, y))\}^2 \\
&= 4\epsilon^2 \|\mathbf{m}_k \odot \nabla_{\mathbf{w}} f(\mathbf{w}; (\mathbf{x}, y))\|_2^2
\end{aligned}$$

Since by Definition 3.1 our sensitive mask would maximize $\|\mathbf{m}_k \odot \nabla_{\mathbf{w}} f(\mathbf{w}; (\mathbf{x}, y))\|^2$ for a given sparsity ratio, we would expect our sensitive mask to *maximize* the magnitude of the loss value difference *for any given sparsity ratio*.

• **Maximum coverage of Hessian diagonal, from the lens of Fisher matrix.**
LLMs are often pre-trained on large text corpus[1] to reach low perplexity before entering the fine-tuning stage. In this case, we would assume $p_{\text{LLM}}(y|\mathbf{x}) \sim p_{\mathcal{D}}(y|\mathbf{x})$, which implies the empirical Fisher $\hat{\mathbf{F}}$ should be close to the (true) Fisher matrix $\mathbf{F}$ as follows:

$$\begin{aligned}
\mathbf{F} &= \mathbb{E}_{\mathbf{x} \sim p_{\mathcal{D}}, \hat{y} \sim p_{\text{LLM}}(\cdot|\mathbf{x})} \nabla_{\mathbf{w}} \log p_{\text{LLM}}(\hat{y}|\mathbf{x})(\nabla_{\mathbf{w}} \log p_{\text{LLM}}(\hat{y}|\mathbf{x}))^\top \\
&\approx \hat{\mathbf{F}} = \mathbb{E}_{(\mathbf{x}, y) \sim p_{\mathcal{D}}} \nabla_{\mathbf{w}} \log p_{\text{LLM}}(y|\mathbf{x})(\nabla_{\mathbf{w}} \log p_{\text{LLM}}(y|\mathbf{x}))^\top
\end{aligned}$$

As we assume the empirical Fisher matrix approximates Fisher, which also approximates the Hessian, and empirical

---

[1] Here we assume data examples $(\mathbf{x}, y) \sim p_{\mathcal{D}}$ in fine-tuning datasets after verbalization would also appear in the large text corpus during pre-training.

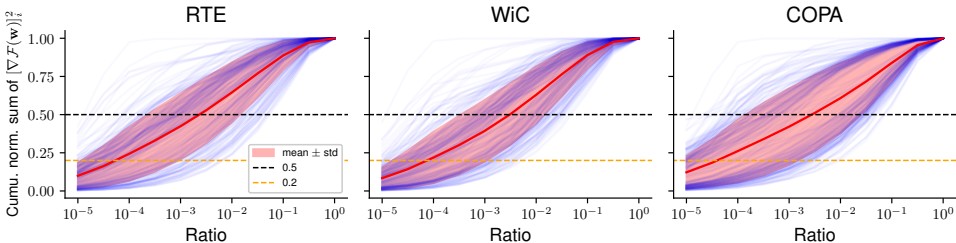

Figure 2: Cumulative normalized sum of coordinate-wise gradient square $[\nabla\mathcal{F}(\mathbf{w})]_i^2$ of linear layers during Llama2-7B (Touvron et al., 2023) fine-tuning. For each linear layer, we first sort parameters by the decreasing order of their gradient square value $[\nabla\mathcal{F}(\mathbf{w})]_i^2, i \in [d_{\text{layer}}]$, and we take the cumulative sum and normalize it to draw a blue curve, and the red-shaded region is the mean $\pm$ std of all blue curves. More similar figures are in Figure 7. **We observe that roughly 0.1% parameters in all linear layers contribute about 50% gradient norm square $\|\nabla\mathcal{F}(\mathbf{w})\|_2^2$.**

Fisher's diagonal is *equal* to the coordinate-wise gradient square vector when computing with downstream task-specific loss, our sensitive parameters would cover a large fraction of the largest Hessian diagonal entries.

This idea of sensitive parameters has been studied in the quantization community (Kim et al., 2023; Guo et al., 2023) and FO optimization (Sung et al., 2021). However, *we are the first one to leverage the extremely sparse sensitive parameters in LLM fine-tuning to accelerate ZO fine-tuning with LLMs*. When we have perturbation and updating in the scale of billion parameters, finding which parameters to fine-tune would be important for improving ZO performance. Notice that here we use sensitive masks $\mathbf{m}_k$ for understanding purposes. In Section 3.4, we will discuss how to transform Definition 3.2 to a parameter-efficient optimization pipeline by optimizing *fixed* sensitive parameters.

### 3.2. Theoretical Convergence Rate

We would investigate the theoretical convergence of sensitive sparse ZO-SGD on sensitive parameters under the non-convex optimization settings. Our assumptions are included in Appendix B.2.

**Theorem 3.3 (Convergence rate of sensitive sparse ZO-SGD (Definition 3.2)).** *If we pick $\eta_t = 1/(L(k+2))$, under Assumptions B.1 (bounded gradient error), B.2 (Lipschitz smoothness), and B.4 (sparse sensitive parameters), we would have*

$$\frac{1}{T}\sum_{t=0}^{T-1}\mathbb{E}_{\bar{\mathbf{z}},(\mathbf{x},y)}\|\nabla_{\mathbf{w}}\mathcal{F}(\mathbf{w}_t)\|^2 \leq$$
$$O\left(\frac{k}{c}\cdot\frac{L}{T}\right)(\mathcal{F}(\mathbf{w}_0) - \mathcal{F}^*) + 3\sigma^2. \quad (5)$$

*Moreover, if we still pick $\eta_t = 1/(L(k+2))$, with an extra Assumption B.3 (P.L. condition), we would have*

$$\mathbb{E}_{\bar{\mathbf{z}},(\mathbf{x},y)}\{\mathcal{F}(\mathbf{w}_T) - \mathcal{F}^*\} \leq$$
$$\left(1 - O\left(\frac{\mu}{L}\cdot\frac{c}{k}\right)\right)^T(\mathcal{F}(\mathbf{w}_0) - \mathcal{F}^*) + \frac{3\sigma^2 c}{2L(k+2)}. \quad (6)$$

The proof for Inequality 5 is in Appendix B.2 and the proof for Inequality 6 is in Appendix B.3. If we choose $k = d$ and $c = 1$, both convergence rates trivially reduce to the standard zeroth-order convergence rate as $O(d/T) + O(\text{constant})$ and $O((1/d)^T) + O(\text{constant})$. As we assume $c \gg k/d$, we know $d \gg k/c$ and therefore both $O((k/c)(1/T))$ and $O((c/k)^T)$ are much lower than $O(d/T) + O(\text{constant})$ and $O((1/d)^T) + O(\text{constant})$ that zeroth-order method will yield.

*We want to emphasize that our contributions are more on empirical LLM fine-tuning instead of general machine learning tasks, and in Section 4.1 we extensively compare our sparse ZO methods with other sparse ZO methods and we demonstrate its superiority during LLM fine-tuning. We do not use the strict "local $r$-effective rank" assumption that Malladi et al. (2023a) uses, and our Assumption B.4 can be easily observed empirically in Figure 2. Liu et al. (2024a) and Ohta et al. (2020) also provide similar analysis on the convergence. However, they do not include our sensitive sparse mask in their studies.*

### 3.3. Transferability of LLM Pre-Training Sparsity Pattern in ZO Fine-Tuning

**Sparse fine-tuning with fixed sensitive parameters.** Our Theorem 3.3 focuses on *dynamic* sparse fine-tuning. However, Panigrahi et al. (2023) notice that in real LLM fine-tuning scenario, the fine-tuning performance could be attributed to a sparse subset of weights ($\sim 0.01\%$). Malladi et al. (2023b) also find certain fine-tuning tasks would demonstrate kernel behaviors, which include "fixed (gradient) features": $\nabla_{\mathbf{w}}f(\mathbf{w}_{\text{after FT}};(\mathbf{x},y)) \sim \nabla_{\mathbf{w}}f(\mathbf{w}_{\text{before FT}};(\mathbf{x},y))$.

The similarity of gradient features during fine-tuning would imply that we *do not* need to re-select our sensitive parameters during fine-tuning i.e. select once *before fine-tuning* should be sufficient. This hypothesis can be validated by Figure 3 and Figure 5b. In Figure 3, the fact that "task grad, static" does *not* vanish and still has a large ratio over "task grad, dyn." at the end of training demonstrate that we can select parameters *before fine-tuning*. We also include similar figures for Mistral-7B and OPT-6.7B in Figure 8 in Appendix C.3. We will describe Figure 5b in Section 4.3.

**Surrogate sensitive sparse mask from pre-training datasets.** Another observation from Figure 3 is that the sensitive parameters derived from pre-training datasets (C4) would still cover a large fraction of model sensitivity. Therefore, we could use it as a *surrogate* sensitive sparse mask when gradients on downstream tasks are unavailable, particularly in scenario of *on-device personalization*. [2]

### 3.4. Our Proposal: ZO LLM Fine-Tuning with Fisher-Informed, Transferable Sparsity

The sparse optimization on *fixed* parameters can be implemented as a parameter-efficient optimization workflow, which will reduce the perturbation and updating time during ZO optimization. Suppose we have derived a sensitive sparse mask $\mathbf{m}_k$, and we know it is fixed during fine-tuning. Instead of applying $\mathbf{m}_k$ to $\mathbf{z}$, we would apply it directly to $\mathbf{w}$ and extract the nonzero parts as below:

$$\mathbf{w}_{\text{sparse}} = \mathbf{w} \odot \mathbf{m}_k, \quad \mathbf{w}_{\text{dense}} = \mathbf{w} \odot (\mathbf{1}_d - \mathbf{m}_k) \quad (7)$$

Denote $\mathbf{z}_{k,t} \sim \mathcal{N}(\mathbf{0}_k, \mathbf{I}_k)$ as the Gaussian perturbation sampled in timestep $t$. We will determine $\mathbf{w}_{\text{sparse}}$ before fine-tuning and optimize on $\mathbf{w}_{\text{sparse}}$ *only* and leave $\mathbf{w}_{\text{dense}}$ frozen during fine-tuning. In this case, our sensitive sparse ZO-SGD update rule will become:

$$\mathbf{w}_{\text{sparse},t+1} = \mathbf{w}_{\text{sparse},t} - \eta_t \hat{g}(\mathbf{w}_{\text{sparse},t}, (\mathbf{x}_t, y_t), \mathbf{z}_{k,t}) \quad (8)$$

In Section 3.5, we will describe how this decomposition would seamlessly combine with existing post-training quantization (PTQ) methods, which creates an opportunity for on-device personalization. In Appendix C.6, we will discuss efficient implementations of linear layers after our decomposition.

---

[2]Obtaining gradients of LLMs on edge devices is expensive, and we usually cannot transfer data from edge devices to the cloud to compute the gradient on downstream tasks on cloud. In this case we would need some surrogate gradient information to derive sensitive sparse masks on cloud. We will discuss this in Section 3.5.

### 3.5. An Opportunity for On-Device LLM Personalization

As LLMs are often pre-trained with user-agnostic public datasets, personalizing LLMs with individual user's preferences and meet user's specific needs before real-world deployment are vital. (Tan et al., 2024b; Mairittha et al., 2020) However, transferring the user-specific data to upstream cloud before fine-tuning LLMs would raise privacy concerns. (Xu et al., 2018) On the other hand, personal devices usually have less computational budget and are more memory-constrained than the cloud (Zhu et al., 2023), and performing full fine-tuning would easily exceed the device memory budget.

If we want to fine-tune a 7B model (e.g., Llama2-7B) on memory-constrained devices, we need to reduce the memory consumption on *model weights*, *gradients*, *forward activations*, and *optimizer states*:

- **Model weights.** We would quantize the $\mathbf{w}_{\text{dense}}$ to 4 bits, which reduces the model size of a Llama-2 7B model from 13.5 to 3.4 GiB.
- **Forward activations.** ZO optimization already saves the need of caching activations.
- **Gradients.** We would use the "random seed trick" same as MeZO (Malladi et al., 2023a) to reproduce layer-wise gradients instead of caching them.
- **Optimizer states.** In this paper we use SGD. Our method can also be implemented as a parameter-efficient optimization method which would also work with other optimizers (even with Adam).

As a result, our memory consumption is *nearly minimum*: we can fine-tune a Llama2-7B model under 8 GiB GPU memory *without any offloading*. This would satisfy the memory constraint by a wide range of edge or mobile devices as illustrated in Table 3.

**Integration with quantization.** In Section 3.4, we know that we can obtain surrogate sensitive sparse masks before fine-tuning. We would first decompose sensitive $\mathbf{w}$ to $\mathbf{w}_{\text{sparse}}$ and $\mathbf{w}_{\text{dense}}$. We will then quantize $\mathbf{w}_{\text{dense}}$. During this process, we will use surrogate gradient information that many PTQ algorithms already have: they need gradients to calibrate their quantization errors.

In addition, our method does *not* put strict constraints on specific choices of quantization algorithms since any algorithm that aims to minimize the least-square quantization error term (as follows) or its variant would suffice: (Chee et al., 2024; Nagel et al., 2020; Frantar et al., 2022; Lin et al., 2023; Kim et al., 2023)

$$Q(\mathbf{w}) = \text{argmin}_{Q(\mathbf{w})} \mathbb{E}_{\mathbf{x}} \|(\mathbf{w} - Q(\mathbf{w}))\mathbf{x}\|_2^2 \quad (9)$$

**On-device personalization workflow.** The workflow is il-

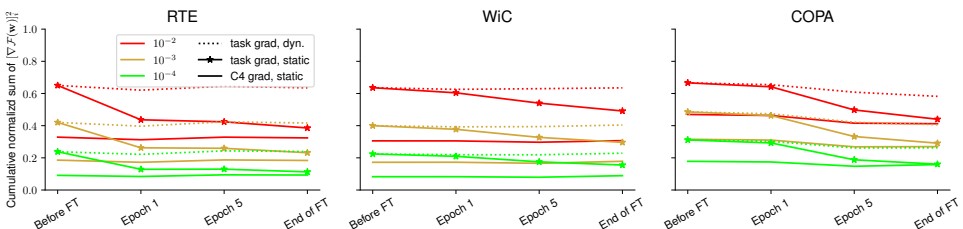

Figure 3: Cumulative normalized gradient square values of Llama2-7B model's linear layers during fine-tuning. For each line, the colors represent the fraction of parameters and the line style represents the category. "task grad, dyn." refers to the sensitive parameters selected at the given timestep (x-axis), and "task grad, static" refers to the sensitive parameters selected before fine-tuning. "C4 grad, static" refers to the sensitive parameters selected with gradients taken from causal language modeling on C4 datasets (Raffel et al., 2019), and we keep it unchanged during fine-tuning. More similar figures are in Figure 8.

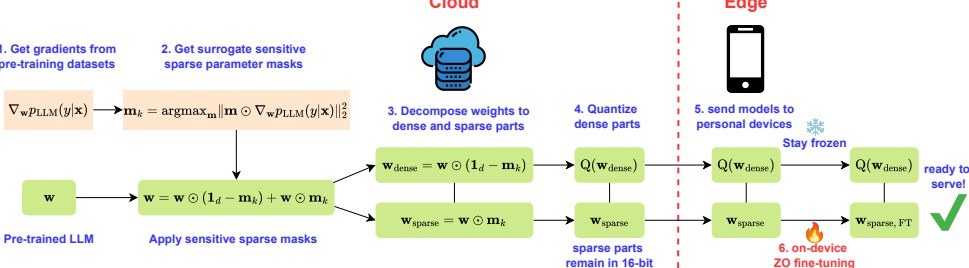

Figure 4: On-device LLM personalization workflow via integrating sensitive sparse ZO optimization with quantization.

lustrated in Figure 4. The high-level overview is that we use surrogate gradient information from pre-training datasets $\nabla_{\mathbf{w}} p_{\text{LLM}}(y|\mathbf{x})$ to extract sensitive parameters $\mathbf{w}_{\text{sparse}}$ and keep $\mathbf{w}_{\text{sparse}}$ in 16 bits, while we quantize the remaining dense weights $\mathbf{w}_{\text{dense}}$ (Step 1-4). We send $\mathbf{w}_{\text{sparse}}$ and $Q(\mathbf{w}_{\text{dense}})$ to personal devices (Step 5), and **we perform on-device ZO fine-tuning only on $\mathbf{w}_{\text{sparse}}$** (Step 6).

## 4. Experiments

In this section, we want to validate the effectiveness of our sensitive sparse ZO optimization method. We also investigate the effectiveness of our on-device personalization recipe in Figure 4. There are a few research questions we want to answer:

- **RQ1:** Is optimizing sensitive parameters more effective than optimizing other subset of parameters during ZO fine-tuning? Can we optimize *surrogate* sensitive sparse parameters when downstream gradient information is unavailable?
- **RQ2:** Can optimizing extremely sparse and *fixed* parameters (Equation 8) lead to iteration-wise and total wall-clock time speedup?
- **RQ3:** Can we match the full performance of ZO full fine-tuning by employing our on-device personalization recipe (Figure 4)?

We focus on 7B-level LLM models (Llama2-7B (Touvron et al., 2023), Mistral-7B (Jiang et al., 2023), OPT-6.7B

(Zhang et al., 2022)) as they would fit with common on-device memory constraints (8 GiB) listed on Table 3 after applying quantization. We use SST-2 (Socher et al., 2013), RTE (Wang et al., 2018), CB (De Marneffe et al., 2019), BoolQ (Clark et al., 2019), WSC (Levesque et al., 2012), WiC (Pilehvar & Camacho-Collados, 2019), and COPA (Roemmele et al., 2011) datasets. We follow standard ZO fine-tuning settings and use the same codebases as in Malladi et al. (2023a). More details of our experiments (hyperparameters, task-specific prompts, etc.) are in Appendix C.

### 4.1. RQ1: Effectiveness of Sparse ZO Fine-Tuning on Sensitive Parameters

We first investigate the performance of optimizing our sensitive parameters versus other subsets of parameters. Our baseline sparsity methods are random subsets and weight outliers. As illustrated in Figure 5a, we can find that ZO fine-tuning would benefit from sparse optimization, as all methods would achieve higher than ZO full fine-tuning at 90% sparsity. However, only sensitive parameters would maintain its performance as we move to the extreme sparsity region ($> 99\%$). *In fact, the performance curve of sensitive parameters w.r.t. different sparsity levels is near a flat curve*, which indicates the performance loss by moving from 90% to 99.9% is minimal. Therefore, we can optimize **100 × less parameters** compared with random and weight outliers and still get same performance.

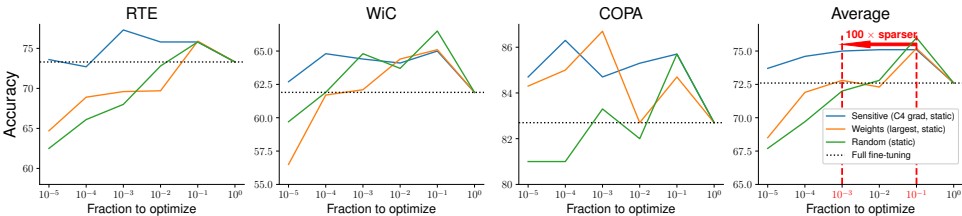

(a) Optimizing sensitive parameters with C4 dataset gradients versus optimizing weights with the largest magnitude (outliers) and random subsets of weights. The trainable parameters are all determined before fine-tuning and other parameters are kept unchanged.

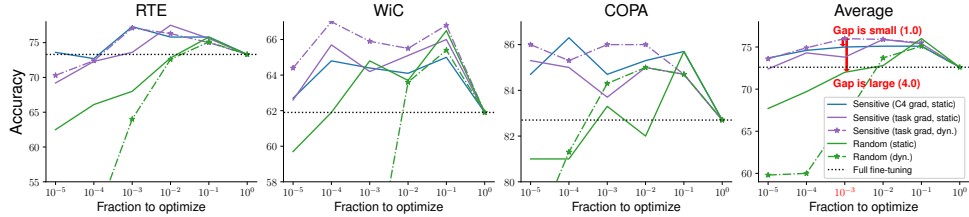

(b) Optimizing sensitive parameters with C4 dataset gradients versus gradients on each fine-tuning task. "Static" means the parameters to optimize are determined before fine-tuning and other parameters are kept unchanged during fine-tuning. "Dyn." means the parameters to optimize will be updated every 100 training steps.

Figure 5: Performance of optimizing sensitive parameters in Llama2-7B fine-tuning on RTE, WiC, and COPA tasks.

We also validate whether optimizing *fixed* and *surrogate* sensitive parameters should still yield satisfactory performance. In Figure 5b, we compare the performance of optimizing sensitive parameters with gradients on C4 dataset with its theoretical upper bound: *fixed* sensitive parameters derived from gradients on each fine-tuning task as the solid line and its dynamic version as the dash-dotted line. We also include the fixed and dynamic random subset parameters as a baseline. We can find that the gap of sensitive parameters between deriving from gradients on C4 dataset and gradients on each fine-tuning task at sparsity level 99.9% is *small* and blue line is still far above the random and full fine-tuning baseline. We also present a summary of our approaches with 99.9% sparsity on various datasets and models in Table 1.

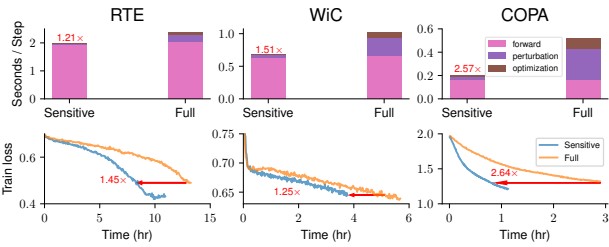

Figure 6: Iteration-wise & wall-clock convergence time of sensitive sparse fine-tuning on fixed parameters ("Sensitive") versus ZO full fine-tuning ("Full") for Llama2-7B. Here we use the 16-bit model as the base model for fine-tuning.

## 4.2. RQ2: Wall-Clock Time Efficiency

By employing parameter-efficient ZO fine-tuning with extreme sparsity, we also achieve 1.2 - 2.5× wall-clock time convergence speedup compared with ZO full fine-tuning as we nearly eliminate the ZO perturbation and optimizer update time, as Figure 6 shows. This also boosts the GPU utilization rate as large-batched ZO forward is often compute-bounded while the perturbation and optimization steps are often memory-bounded. Furthermore, the reduced memory footprint of parameter-efficient ZO fine-tuning allows for training larger models on the same hardware, potentially leading to even better performance. As a result, we answer this question that optimizing extremely sparse and fixed parameters leads to substantial iteration-wise and total wall-clock time improvements.

## 4.3. RQ3: On-Device Personalization

We validate whether our sensitive sparse ZO optimization method would fit with on-device personalization pipeline described in Section 3.5 with Table 1. We follow the exact recipe as described Figure 4 to report a number as "Sensitive (C4, static)", where we only optimize 0.1% sensitive parameters on top of a 4-bit quantized model. As ZO fine-tuning happens *after* model is quantized, ablating on extracting 0.1% random subsets of parameters would produce a *different* quantized model according to Figure 4. So we choose to report the result for optimizing a fixed random subset on top of the 16-bit model as the "Random (static)" as a

Table 1: Performance of difference methods on Llama2-7B fine-tuning tasks. In the first column, "Q" means the full model is quantized with 4-bit quantization method (SqueezeLLM (Kim et al., 2023)), and "ZO" means the model is finetuned with ZO-SGD optimizer. For each cell, we use the same hyperparameters and repeat it with 3 random seeds. We report the average and standard deviation of test set accuracy in the format of **mean**$_{\text{std}}$. In last 2 columns, "Acc" means the average test set accuracy and "Rank" means the average rank of test set accuracy among all methods across datasets.

(a) **Llama2-7B**

|  | Methods | SST-2 | RTE | CB | BoolQ | WSC | WiC | COPA | Acc | Rank |
|---|---|---|---|---|---|---|---|---|---|---|
| Q, ZO | **Sensitive (C4, static)** | **94.7**$_{0.4}$ | **74.7**$_{1.2}$ | **66.7**$_{2.2}$ | **83.0**$_{0.5}$ | 57.4$_{3.9}$ | **65.2**$_{0.9}$ | 85.0$_{2.2}$ | **75.2** | **2.43** |
|  | LoRA | 93.8$_{0.6}$ | 64.7$_{1.1}$ | 64.9$_{4.7}$ | 79.7$_{1.1}$ | **61.5**$_{2.1}$ | 59.8$_{0.1}$ | **85.7**$_{0.5}$ | 72.9 | 4.29 |
|  | Prefix | 80.5$_{4.3}$ | 65.5$_{1.2}$ | 63.1$_{3.0}$ | 80.3$_{0.2}$ | 54.5$_{11.4}$ | 58.3$_{1.3}$ | 82.0$_{0.8}$ | 69.2 | 5.86 |
| ZO | **Sensitive (task, static)** | **94.8**$_{0.1}$ | **73.6**$_{0.9}$ | **69.1**$_{2.2}$ | **83.5**$_{0.8}$ | 57.4$_{4.7}$ | 64.2$_{1.1}$ | **83.7**$_{2.4}$ | **75.2** | **2.29** |
|  | Random (static) | 94.1$_{0.3}$ | 68.0$_{1.7}$ | 64.9$_{3.4}$ | 77.0$_{0.7}$ | **59.6**$_{3.6}$ | **64.8**$_{1.1}$ | 83.3$_{1.7}$ | 73.1 | 4.14 |
|  | Full fine-tuning | 94.6$_{0.5}$ | 73.3$_{5.1}$ | 66.7$_{0.8}$ | 81.9$_{0.8}$ | 58.0$_{4.3}$ | 61.9$_{0.2}$ | 82.7$_{1.7}$ | 74.2 | 3.57 |
|  | Zero-shot | 89.0$_{0.0}$ | 57.8$_{0.0}$ | 32.1$_{0.0}$ | 69.9$_{0.2}$ | 50.2$_{0.0}$ | 36.5$_{0.0}$ | 79.0$_{0.0}$ | 59.2 | 7.29 |
|  | ICL | 94.8$_{0.2}$ | 71.5$_{4.3}$ | 72.6$_{15.2}$ | 77.5$_{4.6}$ | 53.2$_{1.1}$ | 61.1$_{4.3}$ | 87.0$_{2.2}$ | 74.0 | 3.43 |

(b) **Mistral-7B**

|  | Methods | SST-2 | RTE | CB | BoolQ | WSC | WiC | COPA | Acc | Rank |
|---|---|---|---|---|---|---|---|---|---|---|
| Q, ZO | **Sensitive (C4, static)** | 94.0$_{0.3}$ | **74.2**$_{2.7}$ | **70.2**$_{2.2}$ | **75.1**$_{2.4}$ | 59.6$_{4.9}$ | **61.2**$_{0.9}$ | **88.3**$_{1.2}$ | **74.7** | **2.86** |
|  | LoRA | **94.0**$_{0.4}$ | 65.3$_{1.3}$ | 64.9$_{4.5}$ | 70.3$_{3.7}$ | 60.9$_{3.7}$ | 61.1$_{0.4}$ | **88.3**$_{0.5}$ | 72.1 | 3.57 |
|  | Prefix | 86.9$_{2.1}$ | 57.3$_{1.4}$ | 63.7$_{5.9}$ | 62.2$_{0.9}$ | 60.3$_{4.6}$ | 49.0$_{0.3}$ | 81.3$_{1.7}$ | 65.8 | 4.86 |
| ZO | **Sensitive (task, static)** | **94.7**$_{0.3}$ | **77.1**$_{0.9}$ | **69.0**$_{0.8}$ | **78.4**$_{2.2}$ | **58.0**$_{4.3}$ | 61.4$_{0.2}$ | **89.3**$_{1.3}$ | **75.4** | **1.86** |
|  | Random (static) | 87.9$_{1.9}$ | 50.2$_{0.8}$ | 66.1$_{4.4}$ | 60.6$_{1.7}$ | 57.6$_{1.4}$ | 57.3$_{0.8}$ | 82.3$_{1.7}$ | 66.0 | 5.29 |
|  | Full fine-tuning | 94.6$_{0.1}$ | 74.6$_{2.1}$ | 68.8$_{6.2}$ | 76.6$_{0.2}$ | 54.8$_{6.2}$ | **62.6**$_{0.5}$ | 88.3$_{0.5}$ | 74.3 | 2.86 |
|  | Zero-shot | 54.8$_{0.0}$ | 50.5$_{0.0}$ | 37.5$_{0.0}$ | 43.4$_{1.8}$ | 50.8$_{0.0}$ | 39.4$_{0.0}$ | 78.0$_{0.0}$ | 50.6 | 7.00 |
|  | ICL | 60.7$_{16.7}$ | 55.2$_{4.7}$ | 33.3$_{13.1}$ | 46.8$_{6.5}$ | 50.4$_{0.6}$ | 63.8$_{0.9}$ | 88.7$_{0.5}$ | 57.0 | 5.43 |

(c) **OPT-6.7B**

|  | Methods | SST-2 | RTE | CB | BoolQ | WSC | WiC | COPA | Acc | Rank |
|---|---|---|---|---|---|---|---|---|---|---|
| Q, ZO | **Sensitive (C4, static)** | **94.9**$_{0.5}$ | **72.8**$_{3.6}$ | **83.3**$_{5.1}$ | **73.1**$_{0.9}$ | 59.3$_{5.3}$ | **60.9**$_{0.4}$ | **84.0**$_{1.4}$ | **75.5** | **1.29** |
|  | LoRA | 94.2$_{0.2}$ | 69.6$_{1.6}$ | 69.0$_{1.7}$ | 69.6$_{2.0}$ | 57.1$_{9.1}$ | 57.2$_{0.8}$ | 83.0$_{2.2}$ | 71.4 | 4.57 |
|  | Prefix | 93.3$_{0.4}$ | 71.2$_{1.0}$ | 72.0$_{1.7}$ | 68.9$_{2.8}$ | 62.5$_{2.4}$ | 59.4$_{0.5}$ | 80.0$_{2.4}$ | 72.5 | 4.14 |
| ZO | **Sensitive (task, static)** | **94.5**$_{0.4}$ | **75.5**$_{1.4}$ | **82.1**$_{3.6}$ | **72.5**$_{0.8}$ | 57.4$_{5.2}$ | **60.6**$_{1.4}$ | **83.3**$_{1.7}$ | **75.1** | **2.14** |
|  | Random (static) | 87.3$_{2.0}$ | 68.4$_{1.7}$ | 70.6$_{6.3}$ | 66.0$_{1.0}$ | 58.0$_{7.0}$ | 56.4$_{1.3}$ | 79.0$_{0.8}$ | 69.4 | 5.71 |
|  | Full fine-tuning | 94.4$_{0.3}$ | 72.7$_{1.2}$ | 79.8$_{3.0}$ | 72.1$_{1.2}$ | **57.4**$_{4.6}$ | 60.2$_{0.9}$ | 82.3$_{2.6}$ | 74.1 | 3.29 |
|  | Zero-shot | 61.0$_{0.0}$ | 60.7$_{0.0}$ | 46.4$_{0.0}$ | 55.7$_{1.0}$ | 55.5$_{0.0}$ | 36.5$_{0.0}$ | 77.0$_{0.0}$ | 56.1 | 7.71 |
|  | ICL | 74.0$_{14.6}$ | 65.8$_{11.2}$ | 54.8$_{5.9}$ | 67.9$_{2.1}$ | 53.2$_{1.7}$ | 41.0$_{4.5}$ | 80.7$_{2.9}$ | 62.5 | 6.57 |

theoretically performance upper bound.

We also compare with optimizing with LoRA (Hu et al., 2021) and Prefix Tuning (Li & Liang, 2021) with ZO-SGD optimizer on top of the *same* quantized model as "Sensitive (C4, static)". The zero-shot inference and in-context learning (ICL) baselines are also included as the last two rows in each subtable. We follow the LoRA $r$ and $\alpha$ and prefix length shown in Malladi et al. (2023a), and for LoRA, we add it to all linear layers where our sensitive parameters are extracted. We find that integrating sensitive sparse ZO optimization with on-device personalization pipelines would still yield good performance exceeding all baselines across models and tasks. Particularly, the performance is higher than ICL, and ZO full fine-tuning in 16 bits. In addition, we have surpassed other common ZO-PEFT methods and random sparse ZO fine-tuning methods. This demonstrates the superiority of optimizing sensitive parameters *only* in ZO fine-tuning recipes. We also notice that optimizing sensitive parameters derived from C4 gradients still produce close re-

sults as from task-specific gradients (in average less than 1% accuracy difference). This indicates optimizing *surrogate* sensitive parameters is still empirically successful.

## 5. Conclusion

We have shown that the sensitive parameters provided by the pre-training process can effectively assist in ZO LLMs fine-tuning. Our experiments suggest that the ZO fine-tuning guided by 0.1% sensitive parameters in the LLM can even perform better than the full parameter ZO fine-tuning. The experiment results also demonstrate that the quantization of parameters other than sensitive parameters allows us to perform ZO fine-tuning of an LLM on limited memory devices.

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

# Appendix

In Section A we describe all notations used in this paper. In Section B, we include the assumption and exact proof on the convergence rate (Theorem 3.3). In Section C, we describe all details in our experiments and provide a high-level recommendation on how to efficiently implement our sensitive sparse ZO fine-tuning in forward passes of linear layers with existing quantization methods or training / inference workflow.

## A. Notations

We present the notations used in this work as follows.

Table 2: Notations used in this paper

| Term/Symbol | Explanation |
|---|---|
| $f$ | loss function |
| $t$ | optimization timestep $t$ |
| $d$ | number of model parameters |
| $d_{\text{layer}}$ | number of parameters in one linear layer. This means the total number of parameters per each linear layer as the number of rows times the number of columns in each linear layer. |
| $(\mathbf{x}_t, y_t)$ | a data example sampled at timestep $t$ as a pair of input vector and training target |
| $\mathbf{w}_t \in \mathbb{R}^d$ | weight/parameter vector at optimization timestep $t$ |
| $f(\mathbf{w}; (\mathbf{x}, y))$ | training loss of $\mathbf{w}$ evaluated at a single data example $(\mathbf{x}, y)$ |
| $\mathcal{F}(\mathbf{w}) = \mathbb{E}_{(\mathbf{x},y)} f(\mathbf{w}; (\mathbf{x}, y))$ | full-batched training loss of $\mathbf{w}$ |
| $\epsilon$ | a small perturbation scaling constant (close to 0) |
| $\mathbf{z}_t \in \mathbb{R}^d$ | random Gaussian perturbation vector sampled at timestep $t$ |
| $\hat{g}(\mathbf{w}, (\mathbf{x}, y), \mathbf{z})$ | estimated ZO surrogate gradient for $\mathbf{w}$ with a data example $(\mathbf{x}, y)$ and a sampled Gaussian perturbation vector $\mathbf{z}$ (Definition 2.1) |
| $\eta_t$ | learning rate for ZO-SGD optimizer (Definition 2.2) at timestep $t$ |
| $\mathbf{m}_k \in \{0, 1\}^d$ | a sensitive sparse mask with $k$ nonzero entries (Definition 3.1) |
| $\mathbf{m}_{k,t} \in \{0, 1\}^d$ | a sensitive sparse mask with $k$ nonzero entries, and it is derived at optimization timestep $t$ |
| $\mathbf{I}_d$ | Identity matrix with shape $\mathbb{R}^{d \times d}$ |
| $\tilde{\mathbf{I}}_{d, \mathbf{m}_k}$ | $\tilde{\mathbf{I}}_{d, \mathbf{m}_k}$ is equal to the identity matrix $\mathbf{I}_d$ with the main diagonal masked by $\mathbf{m}_k$ |
| $\bar{\mathbf{z}}_t = \mathbf{z}_t \odot \mathbf{m}_k$ | a sampled Gaussian perturbation vector $\mathbf{z}_t$ at timestep $t$ that is masked by $\mathbf{m}_k$. Notice that $\bar{\mathbf{z}}$ is equivalent as being sampled from $\mathcal{N}(\mathbf{0}_d, \tilde{\mathbf{I}}_{d, \mathbf{m}_k})$ |
| $\mathbf{1}_d$ | a vector of size $d$ with all entries equal to 1 |
| Tr | trace operation |
| $Q(\mathbf{w})$ | parameter vector $\mathbf{w}$ that is quantized by $Q$ |
| $\mathbf{F}$ | (true) Fisher information matrix |
| $\hat{\mathbf{F}}$ | empirical Fisher information matrix |
| $p_{\text{LLM}}$ | LLM as a probabilistic model |
| $p_{\mathcal{D}}$ | true data distribution |
| $\mathbf{w}_{\text{sparse}} = \mathbf{w} \odot \mathbf{m}_k$ | sensitive parameters with positions as the nonzero entries sensitive sparse mask $\mathbf{m}_k$ (Equation 7) |
| $L$ | Lipschitz constant in Assumption B.2 |
| $\mu$ | PL condition number in Assumption B.3 |
| $\sigma^2$ | stochastic gradient error term in Assumption B.1 |
| $W_K$ | weight matrix of linear projection for the key embedding matrix $K$ in attention layers |
| $W_V$ | weight matrix of linear projection for the value embedding matrix $V$ in attention layers |

# B. Theoretical Convergence Rate

## B.1. Assumptions

We start with listing standard assumptions in nonconvex optimization literature:

**Assumption B.1** (**Bounded stochastic gradient errors**). For any data example $(\mathbf{x}, y) \in \mathcal{D}$ and for any $\mathbf{w} \in \mathbb{R}^d$, denote the full-batched loss function $\mathcal{F}(\mathbf{w}) = \mathbb{E}_{(\mathbf{x},y) \in \mathcal{D}} f(\mathbf{w}; (\mathbf{x}, y))$, we have

$$\|\nabla_{\mathbf{w}} f(\mathbf{w}; (\mathbf{x}, y)) - \nabla_{\mathbf{w}} \mathcal{F}(\mathbf{w})\|^2 \leq \sigma^2. \tag{10}$$

**Assumption B.2** (**Lipschitz smoothness**). We assume that $f(\mathbf{w}, \mathbf{x})$ is $L$-Lipschitz smooth ($L > 0$): for any $\mathbf{w}, \mathbf{w}' \in \mathbb{R}^d$,

$$\|\nabla_{\mathbf{w}} f(\mathbf{w}; (\mathbf{x}, y)) - \nabla_{\mathbf{w}} f(\mathbf{w}'; (\mathbf{x}, y))\| \leq L\|\mathbf{w} - \mathbf{w}'\|. \tag{11}$$

**Assumption B.3** (**PL inequality**). We assume that $\mathcal{F}(\mathbf{w})$ fulfills the Polyak-Lojasiewicz (PL) condition: there exists some $\mu > 0$, for any $\mathbf{w} \in \mathbb{R}^d$

$$\frac{1}{2}\|\nabla_{\mathbf{w}} \mathcal{F}(\mathbf{w})\|^2 \geq \mu(\mathcal{F}(\mathbf{w}) - \mathcal{F}^*), \quad \mathcal{F}^* \text{ is the minimum value } \mathcal{F}^* = \inf_{\mathbf{w}} \mathcal{F}(\mathbf{w}). \tag{12}$$

Inspired by Figure 7, we would assume the sensitive parameters of $\mathbf{w}$ are sparse.

**Assumption B.4** (**Sensitive parameters are sparse**). We assume at timestep $t$ $\exists \mathbf{m}_t \in \{0,1\}^d$ with the number of nonzero entries as $k$, $\exists c \in [0, 1]$ such that

$$\|\mathbf{m}_t \odot \nabla_{\mathbf{w}} f(\mathbf{w}_t; (\mathbf{x}_t, y_t))\|^2 = c\|\nabla_{\mathbf{w}} f(\mathbf{w}_t; (\mathbf{x}_t, y_t))\|^2.$$

Here we assume $c \gg k/d$. [3]

## B.2. Proof for Equation 5, Theorem 3.3

We will start with formulating the expectation of sensitive sparse ZO surrogate gradient norm square in terms of its corresponding stochastic gradient norm square.

**Lemma B.5** (**Sensitive sparse ZO surrogate gradient norm square**).

$$\mathbb{E}_{\bar{\mathbf{z}}}[\|\hat{g}(\mathbf{w}_t, (\mathbf{x}_t, y_t), \bar{\mathbf{z}}_t)\|^2] = (2 + k)c\|\nabla_{\mathbf{w}} f(\mathbf{w}, (\mathbf{x}_t, y_t))\|^2$$

***Proof for Lemma B.5.*** We know that our $\bar{\mathbf{z}}$ can be considered as being sampled from $\mathcal{N}(\mathbf{0}, \tilde{\mathbf{I}}_{d,\mathbf{m}_k})$ where $\tilde{\mathbf{I}}_{d,\mathbf{m}_k}$ is the identity matrix $\mathbf{I}_d$ with the main diagonal masked by $\mathbf{m}_k$.

We expand the sensitive sparse ZO surrogate gradient covariance matrix $\mathbb{E}_{\bar{\mathbf{z}}} \hat{g}(\mathbf{w}, (\mathbf{x}, y), \bar{\mathbf{z}}) \hat{g}(\mathbf{w}, (\mathbf{x}, y), y), \bar{\mathbf{z}})^\top$ as follows:

$$\begin{aligned}
&\mathbb{E}_{\bar{\mathbf{z}}} \hat{g}(\mathbf{w}, (\mathbf{x}, y), \bar{\mathbf{z}}) \hat{g}(\mathbf{w}, (\mathbf{x}, y), \bar{\mathbf{z}})^\top \\
=& \mathbb{E}_{\bar{\mathbf{z}}_i} [\bar{\mathbf{z}}_i \bar{\mathbf{z}}_i^\top \left( (\mathbf{m}_k \odot \nabla_{\mathbf{w}} f(\mathbf{w}; (\mathbf{x}, y)))(\mathbf{m}_k \odot \nabla_{\mathbf{w}} f(\mathbf{w}; (\mathbf{x}, y)))^\top \right) \bar{\mathbf{z}}_i \bar{\mathbf{z}}_i^\top] \\
=& 2 \left( (\mathbf{m}_k \odot \nabla_{\mathbf{w}} f(\mathbf{w}; (\mathbf{x}, y)))(\mathbf{m}_k \odot \nabla_{\mathbf{w}} f(\mathbf{w}; (\mathbf{x}, y)))^\top \right) + \|\mathbf{m}_k \odot \nabla_{\mathbf{w}} f(\mathbf{w}; (\mathbf{x}, y))\|^2 \tilde{\mathbf{I}}_{d,\mathbf{m}_k}
\end{aligned}$$

Then the sensitive sparse ZO surrogate gradient norm square is the square of the *diagonal* of its corresponding covariance matrix:

$$\begin{aligned}
\mathbb{E}_{\bar{\mathbf{z}}}[\|\hat{g}(\mathbf{w}_t, \mathbf{x}_t, \bar{\mathbf{z}}_t)\|^2] &= \text{diag} \left( \mathbb{E}_{\bar{\mathbf{z}}} \hat{g}(\mathbf{w}, (\mathbf{x}, y), \bar{\mathbf{z}}) \hat{g}(\mathbf{w}, (\mathbf{x}, y), y), \bar{\mathbf{z}})^\top \right)^2 \\
&= 2c\|\nabla_{\mathbf{w}} f(\mathbf{w}, (\mathbf{x}_t, y_t))\|^2 + kc\|\nabla_{\mathbf{w}} f(\mathbf{w}, (\mathbf{x}_t, y_t))\|^2 \\
&= (2 + k)c\|\nabla_{\mathbf{w}} f(\mathbf{w}, (\mathbf{x}_t, y_t))\|^2
\end{aligned}$$

$\square$

---

[3]From Figure 7, we know that for $c \sim 0.5$, we only need $k/d \sim 0.001$. In this case $k/c \sim 0.002d$.

Then we are in good shape of deriving the convergence rate under the Lipschitz smoothness condition:

***Proof for Equation 5, Theorem 3.3.***

$$f(\mathbf{w}_{t+1}, \mathbf{x}_t) \le f(\mathbf{w}_t; (\mathbf{x}_t, y_t)) + \langle \nabla f(\mathbf{w}_t; (\mathbf{x}_t, y_t)), \mathbf{w}_{t+1} - \mathbf{w}_t \rangle + \frac{L}{2} \|\mathbf{w}_{t+1} - \mathbf{w}_t\|^2$$

$$\le f(\mathbf{w}_t; (\mathbf{x}_t, y_t)) - \eta_t \langle \nabla f(\mathbf{w}_t; (\mathbf{x}_t, y_t)), \hat{g}(\mathbf{w}_t, \mathbf{x}_t, \bar{\mathbf{z}}_t) \rangle + \frac{L\eta_t^2}{2} \|\hat{g}(\mathbf{w}_t, \mathbf{x}_t, \bar{\mathbf{z}}_t)\|^2$$

$$\mathbb{E}_{\bar{\mathbf{z}}} f(\mathbf{w}_{t+1}, \mathbf{x}_t) \le \mathbb{E}_{\bar{\mathbf{z}}} f(\mathbf{w}_t; (\mathbf{x}_t, y_t)) - \eta_t \mathbb{E}_{\bar{\mathbf{z}}} \|\mathbf{m}_{k,t} \odot \nabla f(\mathbf{w}_t; (\mathbf{x}_t, y_t))\|^2 + \frac{L\eta_t^2}{2} \mathbb{E}_{\bar{\mathbf{z}}} \|\hat{g}(\mathbf{w}_t, \mathbf{x}_t, \bar{\mathbf{z}})\|^2$$

$$\mathbb{E}_{\bar{\mathbf{z}}} f(\mathbf{w}_{t+1}, \mathbf{x}_t) \le \mathbb{E}_{\bar{\mathbf{z}}} f(\mathbf{w}_t; (\mathbf{x}_t, y_t)) - c\eta_t \mathbb{E}_{\bar{\mathbf{z}}} \|\nabla f(\mathbf{w}_t; (\mathbf{x}_t, y_t))\|^2 + \frac{L\eta_t^2}{2} c(k+2) \mathbb{E}_{\bar{\mathbf{z}}} \|\nabla_{\mathbf{w}} f(\mathbf{w}_t; (\mathbf{x}_t, y_t))\|^2$$

$$\mathbb{E}_{\bar{\mathbf{z}},(\mathbf{x},y)} \mathcal{F}(\mathbf{w}_{t+1}) \le \mathbb{E}_{\bar{\mathbf{z}},(\mathbf{x},y)} \{\mathcal{F}(\mathbf{w}_t) - c\eta_t \|\nabla_{\mathbf{w}} \mathcal{F}(\mathbf{w}_t)\|^2 + c\sigma^2 \eta_t + \frac{L\eta_t^2}{2} c(k+2) \|\nabla_{\mathbf{w}} \mathcal{F}(\mathbf{w}_t)\|^2 + \frac{L\eta_t^2}{2} c(k+2)\sigma^2 \}$$

$$\mathbb{E}_{\bar{\mathbf{z}},(\mathbf{x},y)} \mathcal{F}(\mathbf{w}_{t+1}) \le \mathbb{E}_{\bar{\mathbf{z}},(\mathbf{x},y)} \{\mathcal{F}(\mathbf{w}_t) - \left( c\eta_t - \frac{L\eta_t^2}{2} c(k+2) \right) \|\nabla_{\mathbf{w}} \mathcal{F}(\mathbf{w}_t)\|^2 + \left( c\sigma^2 \eta_t + \frac{L\eta_t^2}{2} c(k+2)\sigma^2 \right) \}$$

Denote $\alpha = Lc(k+2)$, we will have

$$\mathbb{E}_{\bar{\mathbf{z}},(\mathbf{x},y)} \mathcal{F}(\mathbf{w}_{t+1}) \le \mathbb{E}_{\bar{\mathbf{z}},(\mathbf{x},y)} \{\mathcal{F}(\mathbf{w}_t) - \eta_t \left( c - \frac{\alpha}{2}\eta_t \right) \|\nabla_{\mathbf{w}} \mathcal{F}(\mathbf{w}_t)\|^2 + \left( c\sigma^2 \eta_t + \frac{\alpha}{2}\sigma^2 \eta_t^2 \right) \}$$

Set $\eta_t < \frac{c}{\alpha} = \frac{1}{L(k+2)}$, we have

$$\mathbb{E}_{\bar{\mathbf{z}},(\mathbf{x},y)} \mathcal{F}(\mathbf{w}_{t+1}) \le \mathbb{E}_{\bar{\mathbf{z}},(\mathbf{x},y)} \{\mathcal{F}(\mathbf{w}_t) - \frac{c\eta_t}{2} \|\nabla \mathcal{F}(\mathbf{w}_t)\|^2 + \left( c\sigma^2 \eta_t + \frac{\alpha}{2}\sigma^2 \eta_t^2 \right) \}$$

If we apply our sparse ZO update rule recursively for $T$ steps,

$$\frac{1}{T} \sum_{t=0}^{T-1} \mathbb{E}_{\bar{\mathbf{z}},(\mathbf{x},y)} \|\nabla_{\mathbf{w}} \mathcal{F}(\mathbf{w}_t)\|^2 \le \frac{2\alpha}{Tc^2} (\mathcal{F}(\mathbf{w}_0) - \mathcal{F}^*) + \frac{1}{T} \sum_{t=0}^{T-1} \frac{\left( c\sigma^2 \eta_t + \frac{\alpha}{2}\sigma^2 \eta_t^2 \right)}{\frac{c\eta_t}{2}}$$

$$\le \frac{2\alpha}{Tc^2} (\mathcal{F}(\mathbf{w}_0) - \mathcal{F}^*) + (2\sigma^2 + \sigma^2)$$

$$\le \frac{2L(k+2)}{c} \frac{1}{T} (\mathcal{F}(\mathbf{w}_0) - \mathcal{F}^*) + 3\sigma^2$$

$$\le O\left( \frac{k}{c} \cdot \frac{L}{T} \right) (\mathcal{F}(\mathbf{w}_0) - \mathcal{F}^*) + 3\sigma^2$$

□

## B.3. Proof for Equation 6, Theorem 3.3

We can derive a convergence rate of sensitive sparse ZO-SGD optimization method under PL inequality and Lipschitz-smoothness as follows (this proof resumes from our prior proof with the Lipschitz-smoothness condition alone):

***Proof for Equation 6, Theorem 3.3.*** Denote $\kappa$ as the condition number $\kappa = \frac{\mu}{L}$.

$$\mathbb{E}_{\bar{\mathbf{z}},(\mathbf{x},y)}\mathcal{F}(\mathbf{w}_{t+1}) \leq \mathbb{E}_{\bar{\mathbf{z}},(\mathbf{x},y)}\{\mathcal{F}(\mathbf{w}_t) - \frac{c\eta_t}{2}\|\nabla\mathcal{F}(\mathbf{w}_t)\|^2 + \left(c\sigma^2\eta_t + \frac{\alpha}{2}\sigma^2\eta_t^2\right)\}$$

$$\leq \mathbb{E}_{\bar{\mathbf{z}},(\mathbf{x},y)}\{\mathcal{F}(\mathbf{w}_t) - c\mu\eta_t(\mathcal{F}(\mathbf{w}_t) - \mathcal{F}^*) + \left(c\sigma^2\eta_t + \frac{\alpha}{2}\sigma^2\eta_t^2\right)\}$$

$$\mathbb{E}_{\bar{\mathbf{z}},(\mathbf{x},y)}\{\mathcal{F}(\mathbf{w}_{t+1}) - \mathcal{F}^*\} \leq \mathbb{E}_{\bar{\mathbf{z}},(\mathbf{x},y)}\{(\mathcal{F}(\mathbf{w}_t) - \mathcal{F}^*) - c\mu\eta_t(\mathcal{F}(\mathbf{w}_t) - \mathcal{F}^*) + \left(c\sigma^2\eta_t + \frac{\alpha}{2}\sigma^2\eta_t^2\right)\}$$

$$\mathbb{E}_{\bar{\mathbf{z}},(\mathbf{x},y)}\{\mathcal{F}(\mathbf{w}_{t+1}) - \mathcal{F}^*\} \leq \mathbb{E}_{\bar{\mathbf{z}},(\mathbf{x},y)}\{(\mathcal{F}(\mathbf{w}_t) - \mathcal{F}^*) - c\mu\eta_t(\mathcal{F}(\mathbf{w}_t) - \mathcal{F}^*) + \left(c\sigma^2\eta_t + \frac{\alpha}{2}\sigma^2\eta_t^2\right)\}$$

Plugging in $\eta_t \leq \dfrac{c}{\alpha}$ and applying recursively for $T$ iterations.

$$\mathbb{E}_{\bar{\mathbf{z}},(\mathbf{x},y)}\{\mathcal{F}(\mathbf{w}_T) - \mathcal{F}^*\} \leq (1 - \frac{c\kappa}{(k+2)})^T(\mathcal{F}(\mathbf{w}_0) - \mathcal{F}^*) + \frac{3\sigma^2 c^2}{2\alpha}$$

$$\leq (1 - \frac{c\kappa}{(k+2)})^T(\mathcal{F}(\mathbf{w}_0) - \mathcal{F}^*) + \frac{3\sigma^2 c}{2L(k+2)}$$

$$\leq \left(1 - O\left(\frac{\mu}{L}\cdot\frac{c}{k}\right)\right)^T(\mathcal{F}(\mathbf{w}_0) - \mathcal{F}^*) + \frac{3\sigma^2 c}{2L(k+2)}$$

$\square$

## C. Supplementary Experiment Details

### C.1. On-device memory constraints

We include a table of common memory constraints imposed by edge or mobile devices as Table 3. We can find that a wide range of these devices impose a memory constraint of **8 GiB** as our main main constraint that we consider when we develop our on-device personalization recipe in Section 3.5.

Table 3: Device memory of some mobile devices or consumer-graded GPUs.

| Devices | Memory |
|---|---|
| Nvidia GeForce GTX 1080 Ti | 11 GiB |
| Nvidia GeForce RTX 3060 Ti | 8 GiB |
| Nvidia Jetson TX2 | 8 GiB |
| OPPO Find X7 Ultra (Li et al., 2024) | 12 GiB |
| Samsung Galaxy S10 with Mali-G76 GPU (Gim & Ko, 2022) | 8 GiB |

### C.2. Gradient sparsity during LLM fine-tuning

In Figure 2, we explore the FO gradient sparsity of Llama2-7B during fine-tuning (at Epoch 5). Here we follow the identical setting and plot the FO gradient sparsity for Llama2-7B, Mistral-7B, and OPT-6.7B during epoch 1, 5, and 10 (end of fine-tuning).

We observe that the gradient sparsity is exhibited throughout the fine-tuning with slightly increasing towards the end. OPT-6.7B which uses ReLU as the activation function would demonstrate greater sparsity across tasks compared with Llama2-7B and Mistral-7B which uses SwiGLU and SiLU respectively. Nevertheless, the gradient sparsity pattern holds across architectures, tasks, and fine-tuning time in general.

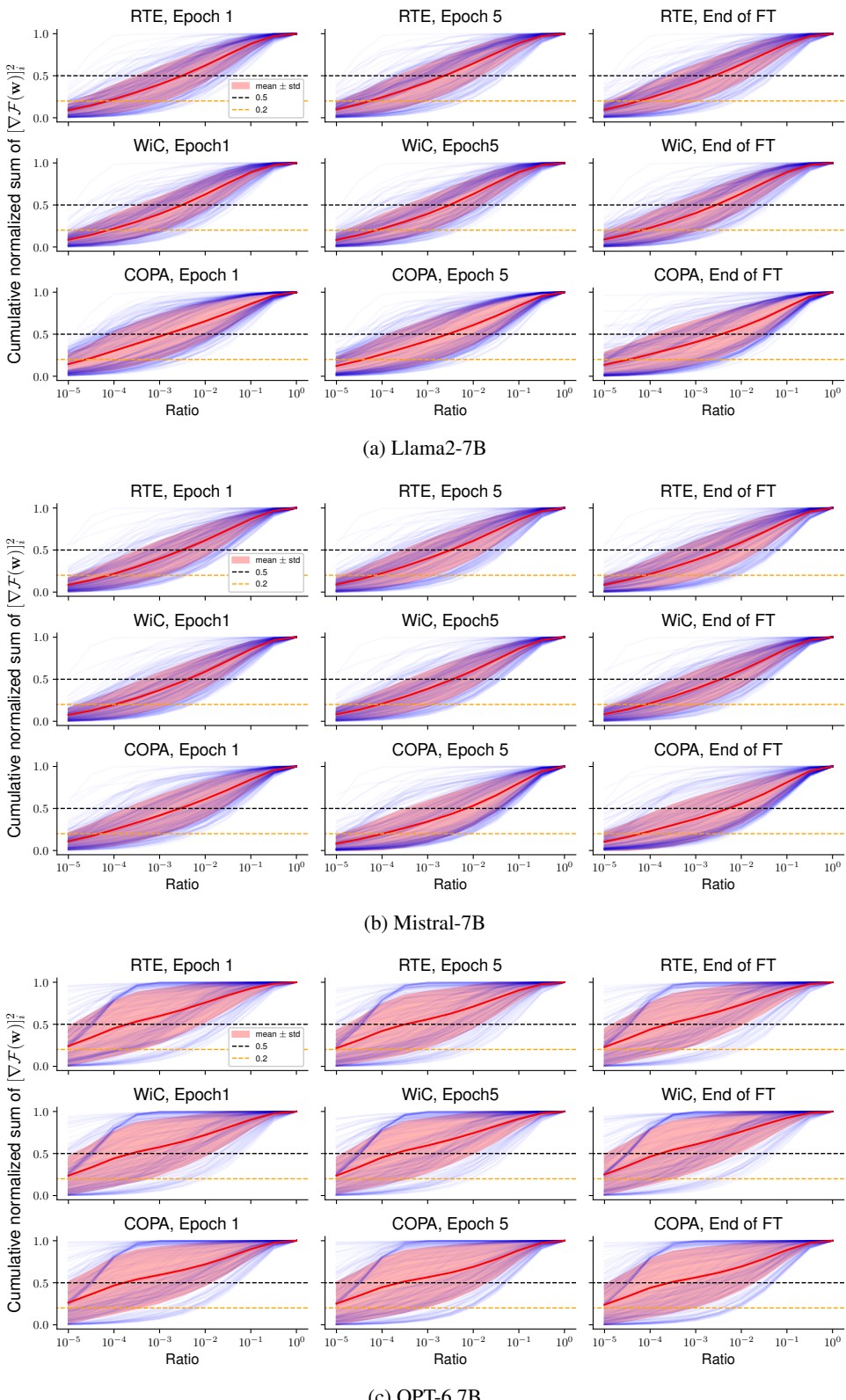

(a) Llama2-7B

(b) Mistral-7B

(c) OPT-6.7B

Figure 7: Cumulative normalized sum of coordinate-wise gradient square $[\nabla \mathcal{F}(\mathbf{w})]_i^2$ of linear layers for Llama2-7B (subfigure 7a), Mistral-7B (subfigure 7b), and OPT-6.7B (subfigure 7c) across RTE, WiC, and COPA tasks during FO-SGD fine-tuning. For each linear layer, we first sort parameters by the decreasing order of their gradient square value $[\nabla \mathcal{F}(\mathbf{w})]_i^2, i \in [d_{\text{layer}}]$, and we take the cumulative sum and normalize it to draw a blue curve, and the red-shaded region is the mean $\pm$ std of all blue curves.

## C.3. Transferability of gradient features from pre-training datasets to downstream tasks

In Figure 3, we explore the transferability of gradient features from pre-training datasets (C4) to downstream tasks, and here we will also validate this phenomenon across models, as shown in Figure 8. As there are *no* solid lines (top-(1e-2,1e-3,1e-4)) parameters with C4 gradient entries prior to fine-tuning) vanish to 0, we know the transferability of gradient features from C4 datasets to downstream datasets hold across models and downstream tasks. In this case, sensitive parameters determined from C4 gradients would still be similar to sensitive parameters determined from downstream task-specific gradients across models.

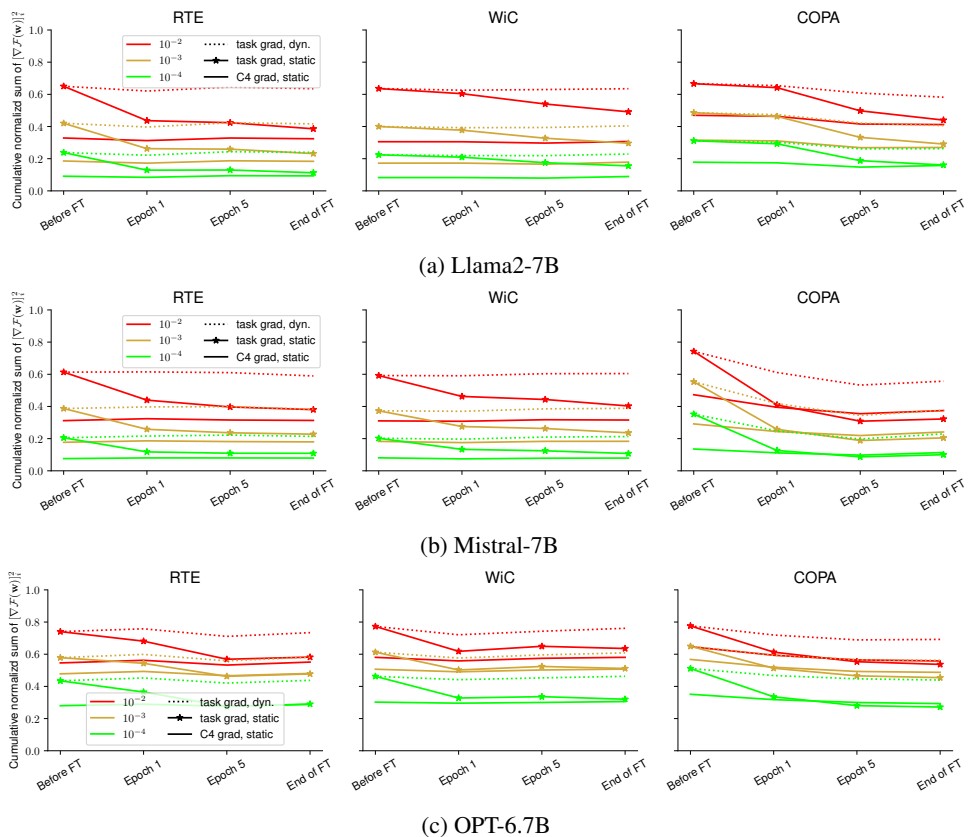

(a) Llama2-7B

(b) Mistral-7B

(c) OPT-6.7B

Figure 8: Cumulative normalized gradient square values of Llama2-7B (subfigure 8a), Mistral-7B (subfigure 8b), and OPT-6.7B (subfigure 8c)'s linear layers during FO fine-tuning. For a given model and training checkpoint, we report the average value across all linear layers as a line in each subfigure. For each line, the colors represent the fraction of parameters (1e-2,1e-3,1e-4) and the line style represents the category. "task grad, dyn." refers to the sensitive parameters selected at the given timestep (x-axis), and "task grad, static" refers to the sensitive parameters selected before fine-tuning. "C4 grad, static" refers to the sensitive parameters selected with gradients taken from causal language modeling on C4 datasets, and we keep it unchanged during fine-tuning.

## C.4. Hyperparameters in experiments

For all experiments, we use 20,000 training steps with ZO-SGD optimizer (Definition 2.2). We will save a model checkpoint every 500 steps, and load the checkpoint with the lowest loss on the validation set at the end of the training, and report its test set accuracy as result. Usually, the training/validation set will be sampled from the original dataset with size 1000/500 respectively and the test set is of size $\min(1000, |\text{original test set}|)$, except for CB and COPA that we use 100 for the validation set size. For all ZO experiments (Table 4 and Table 5), we use batch size of 16. This experiment setting is identical to Malladi et al. (2023a).

Table 4: The chosen hyperparameters for experiments in Table 1. We repeat each hyperparameters for 3 random trials and report the average and standard deviation in Table 1.

(a) **Llama2-7B**

|  | Methods | SST-2 | RTE | CB | BoolQ | WSC | WiC | COPA |
|---|---|---|---|---|---|---|---|---|
| Q, ZO | **Sensitive (C4, static)** ($\epsilon$ =1e-3) | 5e-7 | 1e-6 | 1e-6 | 1e-6 | 5e-7 | 1e-6 | 1e-6 |
|  | LoRA ($\epsilon$ =1e-3) | 1e-5 | 5e-5 | 1e-5 | 2e-5 | 1e-5 | 2e-5 | 1e-5 |
|  | Prefix ($\epsilon$ =1e-2) | 1e-4 | 2e-4 | 5e-4 | 5e-4 | 1e-4 | 5e-4 | 2e-4 |
| ZO | **Sensitive (task, static)** ($\epsilon$ =1e-3) | 5e-7 | 1e-6 | 1e-6 | 1e-6 | 1e-6 | 1e-6 | 2e-6 |
|  | Random (static) ($\epsilon$ =1e-3) | 2e-4 | 5e-4 | 2e-4 | 5e-4 | 2e-4 | 5e-4 | 5e-4 |
|  | Full fine-tuning ($\epsilon$ =1e-3) | 5e-7 | 5e-7 | 5e-7 | 5e-7 | 2e-7 | 5e-7 | 5e-7 |
|  | ICL (#examples) | 16 | 16 | 16 | 8 | 16 | 8 | 8 |

(b) **Mistral-7B**

|  | Methods | SST-2 | RTE | CB | BoolQ | WSC | WiC | COPA |
|---|---|---|---|---|---|---|---|---|
| Q, ZO | **Sensitive (C4, static)** ($\epsilon$ =1e-4) | 2e-8 | 5e-8 | 2e-8 | 2e-8 | 1e-8 | 2e-8 | 2e-8 |
|  | LoRA ($\epsilon$ =1e-4) | 2e-6 | 5e-6 | 2e-6 | 2e-6 | 2e-6 | 2e-6 | 2e-6 |
|  | Prefix ($\epsilon$ =1e-3) | 1e-3 | 2e-3 | 1e-3 | 1e-2 | 5e-4 | 1e-3 | 5e-4 |
| ZO | **Sensitive (task, static)** ($\epsilon$ =1e-4) | 5e-8 | 5e-8 | 2e-8 | 2e-8 | 2e-8 | 2e-8 | 2e-8 |
|  | Random (static) ($\epsilon$ =1e-4) | 1e-5 | 2e-6 | 5e-6 | 1e-5 | 1e-6 | 2e-6 | 2e-5 |
|  | Full fine-tuning ($\epsilon$ =1e-4) | 2e-8 | 2e-8 | 1e-8 | 1e-8 | 1e-8 | 1e-8 | 2e-8 |
|  | ICL (#examples) | 4 | 8 | 4 | 16 | 4 | 4 | 8 |

(c) **OPT-6.7B**

|  | Methods | SST-2 | RTE | CB | BoolQ | WSC | WiC | COPA |
|---|---|---|---|---|---|---|---|---|
| Q, ZO | **Sensitive (C4, static)** ($\epsilon$ =1e-3) | 2e-7 | 5e-7 | 5e-7 | 5e-7 | 2e-7 | 5e-7 | 2e-7 |
|  | LoRA ($\epsilon$ =1e-3) | 1e-5 | 2e-5 | 1e-5 | 2e-5 | 1e-5 | 2e-5 | 2e-5 |
|  | Prefix ($\epsilon$ =1e-2) | 2e-3 | 1e-2 | 1e-3 | 5e-3 | 5e-3 | 1e-2 | 5e-3 |
| ZO | **Sensitive (task, static)** ($\epsilon$ =1e-3) | 2e-7 | 5e-7 | 5e-7 | 2e-7 | 2e-7 | 5e-7 | 2e-7 |
|  | Random (static) ($\epsilon$ =1e-3) | 1e-4 | 5e-5 | 2e-5 | 5e-5 | 2e-4 | 5e-5 | 5e-5 |
|  | Full fine-tuning ($\epsilon$ =1e-3) | 2e-7 | 2e-7 | 2e-7 | 2e-7 | 2e-7 | 2e-7 | 5e-7 |
|  | ICL (#examples) | 16 | 4 | 16 | 16 | 16 | 8 | 16 |

Our hyperparameters (learning rate $\eta$, perturbation scaling constant $\epsilon$, and the number of ICL examples) for Table 1 is reported in Table 4 for reproducibility. We use constant $\eta$ and $\epsilon$ throughout our experiments. We also report the chosen hyperparameter for Figure 5a and Figure 5b in Table 5. For LoRA, we always add to all linear layers with $r = 8$ and $\alpha = 16$, and for Prefix Tuning, we always add to $W_K$ and $W_V$ with length as 5, as what Malladi et al. (2023a) uses.

Table 5: The chosen hyperparameters for experiments in Figure 5a and Figure 5b. We repeat each hyperparameters for 3 random trials and report the average to draw a line in Figure 5a and Figure 5b, and we use Llama2-7B for all experiments. For each subtable, we include the fraction to optimize on its header and report the chosen learning rate on each cell.

(a) **RTE**

| Methods | 1e-5 | 1e-4 | 1e-3 | 1e-2 | 1e-1 |
|---|---|---|---|---|---|
| **Sensitive (C4, static)** ($\epsilon$ =1e-3) | 1e-5 | 1e-6 | 1e-6 | 1e-6 | 1e-6 |
| **Sensitive (task-specific, static)** ($\epsilon$ =1e-3) | 1e-5 | 1e-6 | 1e-6 | 1e-6 | 1e-6 |
| **Sensitive (task-specific, dynamic)** ($\epsilon$ =1e-3) | 1e-5 | 1e-6 | 1e-6 | 1e-6 | 1e-6 |
| Random (static) ($\epsilon$ =1e-3) | 2e-2 | 5e-3 | 5e-4 | 5e-5 | 5e-5 |
| Random (dynamic) ($\epsilon$ =1e-3) | 2e-2 | 5e-3 | 2e-4 | 5e-5 | 5e-6 |
| Weight outliers (static) ($\epsilon$ =1e-3) | 2e-3 | 1e-3 | 2e-4 | 5e-5 | 1e-5 |

(b) **WiC**

| Methods | 1e-5 | 1e-4 | 1e-3 | 1e-2 | 1e-1 |
|---|---|---|---|---|---|
| **Sensitive (C4, static)** ($\epsilon$ =1e-3) | 1e-5 | 2e-6 | 1e-6 | 1e-6 | 1e-6 |
| **Sensitive (task-specific, static)** ($\epsilon$ =1e-3) | 1e-5 | 2e-6 | 1e-6 | 1e-6 | 1e-6 |
| **Sensitive (task-specific, dynamic)** ($\epsilon$ =1e-3) | 1e-5 | 2e-6 | 1e-6 | 1e-6 | 1e-6 |
| Random (static) ($\epsilon$ =1e-3) | 2e-2 | 5e-3 | 5e-4 | 5e-5 | 5e-6 |
| Random (dynamic) ($\epsilon$ =1e-3) | 2e-2 | 5e-3 | 5e-4 | 5e-5 | 5e-6 |
| Weight outliers (static) ($\epsilon$ =1e-3) | 1e-3 | 5e-4 | 2e-4 | 1e-4 | 2e-5 |

(c) **COPA**

| Methods | 1e-5 | 1e-4 | 1e-3 | 1e-2 | 1e-1 |
|---|---|---|---|---|---|
| **Sensitive (C4, static)** ($\epsilon$ =1e-3) | 5e-6 | 1e-6 | 1e-6 | 1e-6 | 5e-7 |
| **Sensitive (task-specific, static)** ($\epsilon$ =1e-3) | 5e-6 | 2e-6 | 2e-6 | 1e-6 | 1e-6 |
| **Sensitive (task-specific, dynamic)** ($\epsilon$ =1e-3) | 5e-6 | 1e-6 | 1e-6 | 1e-6 | 1e-6 |
| Random (static) ($\epsilon$ =1e-3) | 1e-2 | 2e-3 | 5e-4 | 5e-5 | 5e-6 |
| Random (dynamic) ($\epsilon$ =1e-3) | 2e-3 | 1e-3 | 2e-4 | 2e-5 | 2e-6 |
| Weight outliers (static) ($\epsilon$ =1e-3) | 1e-3 | 5e-4 | 5e-4 | 1e-4 | 1e-5 |

## C.5. Task-specific prompts in experiments

We describe our task templates in Table 6.

Table 6: Task templates for all experiments. On the left column we include the task name and the model name, and on the right column we describe the exact prompt with answer candidates.

| Task | Prompts |
|---|---|
| SST-2
(Llama2-7B) | ### Sentence: \<text\> ### Sentiment: negative/positive |
| SST-2
(Mistral-7B, OPT-6.7B) | \<text\> It was terrible/great |
| RTE
(Llama2-7B) | Suppose "\<premise\>" Can we infer that "\<hypothesis\>"? Yes or No?
Yes/No |
| RTE
(Mistral-7B, OPT-6.7B) | \<premise\>
Does this mean that "\<hypothesis\>" is true? Yes or No?
Yes/No |
| CB
(Llama2-7B, Mistral-7B, OPT-6.7B) | Suppose \<premise\> Can we infer that "\<hypothesis\>"? Yes, No, or Maybe?
Yes/No/Maybe |
| BoolQ
(Llama2-7B) | \<passage\> \<question\>? Yes/No |
| BoolQ
(Mistral-7B, OPT-6.7B) | \<passage\> \<question\>?
Yes/No |
| WSC
(Llama2-7B, Mistral-7B, OPT-6.7B) | \<text\>
In the previous sentence, does the pronoun "\<span2\>" refer to \<span1\>? Yes or No?
Yes/No |
| WiC
(Llama2-7B, Mistral-7B, OPT-6.7B) | Does the word "\<word\>" have the same meaning in these two sentences? Yes, No?
\<sent1\>
\<sent2\>
Yes/No |
| COPA
(Llama2-7B, Mistral-7B, OPT-6.7B) | \<premise\> so/because \<candidate\> |

## C.6. Implementation of sparse operations in linear layers

Linear layers in LLMs often contribute most parameters (Kaplan et al., 2020). Since from Equation 7 we know

$$\mathbf{w}_{\text{sparse}} = \mathbf{w} \odot \mathbf{m}_k, \quad \mathbf{w}_{\text{dense}} = \mathbf{w} \odot (\mathbf{1}_d - \mathbf{m}_k), \quad \mathbf{w} = \mathbf{w}_{\text{sparse}} + \mathbf{w}_{\text{dense}} \tag{13}$$

and since $\mathbf{w}_{\text{dense}}$ would have the same shape (and the same computational intensities) as $\mathbf{w}$, we need to improve *wall-clock time* efficiency of $\mathbf{w}_{\text{sparse}}\mathbf{x}$ to improve the computational efficiency of linear layers after extracting the sparse parameters. In this case, we would have two different methods to implement the forward pass of linear layers (with induced sparse operation colored in red):

$$
\begin{aligned}
\mathbf{w}\mathbf{x} &= \mathbf{w}_{\text{dense}}\mathbf{x} + \mathbf{w}_{\text{sparse}}\mathbf{x} & & \tag{14}\\
&= \text{SparseAddMM}(\text{DenseMM}(\mathbf{w}_{\text{dense}}, \mathbf{x}), \mathbf{w}_{\text{sparse}}, \mathbf{x}) & \text{faster with token generation} & \tag{15}\\
&= (\mathbf{w}_{\text{dense}} + \mathbf{w}_{\text{sparse}})\mathbf{x} & & \tag{16}\\
&= \text{DenseMM}(\text{SparseAdd}(\mathbf{w}_{\text{sparse}}, \mathbf{w}_{\text{dense}}), \mathbf{x}) & \text{faster with ZO training} & \tag{17}
\end{aligned}
$$

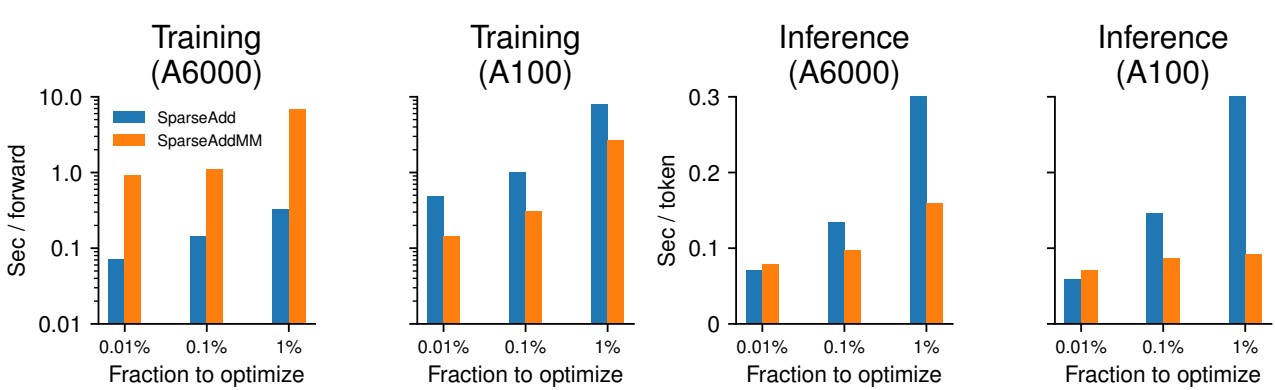

Figure 9: Time of SparseAdd (Equation 17) versus SparseAddMM (Equation 15) in Llama2-7B ZO training forward & inference. In subfigure 1 and 3, we use Nvidia RTX A6000 and Intel Xeno Gold 6342 CPUs, with PyTorch version 2.2, HuggingFace version 4.36, and CUDA 12.2. In subfigure 2 and 4, we use Nvidia A100-SXM4 (40 GiB) and AMD EPYC 7543P 32-Core CPU with PyTorch version 2.1, HuggingFace version 4.38.2, and CUDA 12.2. We use Flash Attention 2 (Dao, 2023) for all 4 subfigures.

The specific choice of employing Equation 15 or Equation 17 needs careful consideration and benchmarking, but here we can provide a general guideline based on the size of input vector (or arithmetic intensity) and potential integration with weight quantization method:

**Size of input vectors $\mathbf{x}$ and arithmetic intensity.** $\mathbf{w}_{\text{sparse}}\mathbf{x}$ in Equation 15 would have a computational dependency over $\mathbf{x}$. During large-batched ZO training, $\mathbf{x}$ would be large enough such that Equation 15 would induce large computational overhead, as shown in subfigure 1 of Figure 9. In contrast, the computational complexity of Equation 17 is *independent* of $\mathbf{x}$ and when $\mathbf{x}$ is large, we would expect Equation 17 is *much* faster than Equation 15. As an example, we use sequence length of 512 and batch size 16 sampled from WikiText-2 dataset (Merity et al., 2016) as a representative computational intensity for ZO training in subfigures 1 and 2 in Figure 9.

However, during autoregressive token generation, on each step we would only append *a single token* to the previously cached embeddings, and in this case $\mathbf{x}$ is small and computing $\mathbf{w}_{\text{dense}} + \mathbf{w}_{\text{sparse}}$ is generally not worthwhile, especially given that $\mathbf{w}_{\text{sparse}}$ is already sparse. This is also illustrated in subfigure 3 and 4 in Figure 9. However, we note that the specific implementation choice is hardware and task dependent and requires thorough benchmarking and we will leave it as a future work.

**We recommend using Equation 17 during large-batched ZO training and Equation 15 during small-batched autoregressive token generation.**

In light of this observation, in our Figure 1, we implement both "SparseAdd" and "SparseAddMM" methods for "Sensitive (0.1%)" and "Random (10%)". For each method we report the *lowest* time out of these 2 implementations: for "Sensitive (0.1%)" training and "Random (10%)" training and inference, we use "SparseAdd" approach. For "Sensitive (0.1%)" inference, we use the "SparseAddMM" approach.

**Integration with weight quantization method.** Weight quantization algorithms can be categorized into 2 categories: uniform quantization method and non-uniform quantization method. For uniform quantization method, (Xi et al., 2023) indicates that we could use integer matrix multiplication to compute $Q(\mathbf{w}_{\text{dense}})\mathbf{x}$ efficiently *without* first dequantizing $Q(\mathbf{w}_{\text{dense}})$ to 16 bits. However, this creates difficulty on our "SparseAdd" approach as we will *violate the constraint of uniformly-spaced quantization bins* by computing $\text{SparseAdd}(Q(\mathbf{w}_{\text{dense}}) + \mathbf{w}_{\text{sparse}})$. In this case, we also have 3 different implementations:

$$Q(\mathbf{w})\mathbf{x} \sim Q(\mathbf{w}_{\text{dense}})\mathbf{x} + \mathbf{w}_{\text{sparse}}\mathbf{x} \tag{18}$$

$$= \text{SparseAddMM}\Big(\text{Dequantize}\big(\text{IntMM}(Q(\mathbf{w}_{\text{dense}}), \mathbf{x})\big), \mathbf{w}_{\text{sparse}}, \mathbf{x}\Big) \quad \text{fits with integer matmul} \tag{19}$$

$$= \text{SparseAddMM}\Big(\text{Dequantize}(Q(\mathbf{w}_{\text{dense}})), \mathbf{x}, \mathbf{w}_{\text{sparse}}\Big) \quad \text{similar to Equation 15} \tag{20}$$

$$= (\text{Dequantize}(Q(\mathbf{w}_{\text{dense}})) + \mathbf{w}_{\text{sparse}})\mathbf{x} \tag{21}$$

$$= \text{DenseMM}(\text{SparseAdd}(\mathbf{w}_{\text{sparse}}, \text{Dequantize}(Q(\mathbf{w}_{\text{dense}})), \mathbf{x}) \quad \text{similar to Equation 17} \tag{22}$$

Equation 19 would compute $\text{IntMM}(Q(\mathbf{w}_{\text{dense}}), \mathbf{x})$ *before* dequantizing it to 16 bits. This would make "SparseAdd" approach infeasible and we can only employ "SparseAddMM" approach in this case. Notice that both Equation 20 and Equation 22 would still dequantize $Q(\mathbf{w}_{\text{dense}})$ first and the choice of implementation would follow into our discussion of input vector size $\mathbf{x}$ in last paragraph. We leave a practical implementation and thorough benchmarking into a future work.

**We recommend using Equation 19 when we use efficient integer matmul to compute $Q(\mathbf{w}_{\text{dense}})\mathbf{x}$ and in other cases, using Equation 20 or Equation 22 follows our previous recommendation based on the size of input vectors.**

### C.7. Hardware, platform, libraries, and other details for fine-tuning and benchmarking

Figure 1, Figure 6, and Figure 9 (subfigure 1 and 3) are trained and evaluated on an internal cluster with 8 Nvidia RTX A6000 GPUs and 2 Intel Xeon Gold 6342 CPUs, with PyTorch version 2.2, HuggingFace version 4.36, and CUDA 12.2. In subfigure 2 and 4 in Figure 9, we use Nvidia A100-SXM4 (40 GiB) and AMD EPYC 7543P 32-Core CPU with PyTorch version 2.1, HuggingFace version 4.38.2, and CUDA 12.2. We use Flash Attention 2 (Dao, 2023) in HuggingFace Transformers library throughout our experiments, and the base model for ZO full fine-tuning and benchmarking is always Llama2-7B with Float16 datatype (torch.float16). We also use the Float16 datatype (torch.float16) for all of our sparse parameters (sensitive sparse, random subsets, etc.) in ZO fine-tuning experiments. Notice that for all of the FO fine-tuning demonstrations (Figure 7 and Figure 8) we use the BrainFloat16 datatype (torch.bfloat16) to avoid the NaN issue from the Float16 datatype.

In Figure 1 and Figure 9, we use sequence length of 512 and batch size 16 sampled from WikiText-2 dataset (Merity et al., 2016) as a representative computational intensity for ZO training, and for inference we generate 128 tokens with top-$p$ ($p = 0.9$) sampling from the prompt "*Please describe the effect of sparse zeroth-order optimization methods on memory-efficient LLM fine-tuning:* ". We still use the Float16 datatype (torch.float16) for both benchmarks.

