# OpenReview forum: "Zeroth-Order Fine-Tuning of LLMs with Extreme Sparsity"
_ICML.cc/2024/Workshop/WANT — WANT@ICML 2024 Poster_

### Official Review · Reviewer_Q9VR · 2024-06-14
**Presents a novel and efficient method for fine-tuning large language models using zeroth-order optimization with extreme sparsity and quantization,**

**Confidence:** 5

**Summary:**

The paper proposes a novel method for fine-tuning large language models (LLMs) using zeroth-order optimization (ZO), which only requires forward passes and is thus memory-efficient. The method integrates sparsity and quantization to enable fine-tuning on memory-constrained devices. The approach identifies and fine-tunes only 0.1% of the most sensitive parameters, achieving performance comparable to full fine-tuning while reducing memory usage and computational requirements. The authors introduce a method to fine-tune LLMs using ZO optimization combined with extreme sparsity and quantization. They show the identification of sensitive parameters that guide efficient ZO fine-tuning. Also, they demonstrate the feasibility of fine-tuning LLMs on devices with limited memory by quantizing non-sensitive parameters. Finally, extensive experiments showing the method's competitive performance across various LLMs and tasks.

**Strengths:**

* The paper presents a novel integration of ZO optimization with extreme sparsity and quantization, which has not been extensively explored in the context of LLM fine-tuning.
* The approach is innovative in addressing memory constraints on edge devices, a growing area of interest as LLMs become more ubiquitous.
* The methods and algorithms proposed are well-justified and grounded in theoretical principles.
* Theoretical claims are supported by both theoretical analysis and empirical evidence, though some assumptions (e.g., sensitive parameter selection) might benefit from further empirical validation.
* The extensive experimental validation demonstrates the method's effectiveness across various models and tasks.
* The paper is well-written and generally easy to follow, with clear motivations and methodology.
* Figures and tables are well-presented and support the text effectively.
* The potential impact on the field is significant, particularly for deploying LLMs on edge devices with constrained resources.
* Contributions are important for advancing the state-of-the-art in efficient LLM fine-tuning and on-device personalization.
* Practical implications are well-discussed, though real-world applications and deployments would further illustrate the method's effectiveness.
* The experiments appear reproducible based on the provided information, but access to the datasets and code would enhance transparency.

**Weaknesses:**

* The paper could provide more detailed supplementary materials to facilitate replication efforts by other researchers.
* The experiments are comprehensive, though more diverse datasets and LLM architectures could strengthen the findings.
* Some sections could benefit from additional explanations, particularly around the theoretical foundations of ZO optimization.
* Practical implications are well-discussed, though real-world applications and deployments would further illustrate the method's effectiveness.
* Some theoretical aspects, particularly around sensitive parameter selection, could benefit from further empirical validation.

**Limitations:**

* The proposed method has strong potential for practical applications, particularly in the deployment of LLMs on mobile and edge devices.
* Conduct a detailed analysis of the method's scalability, particularly in terms of computational and memory requirements. Discuss how the approach scales with different model sizes and how it can be adapted for even larger models or more constrained environments.

**Suggestions:**

* Future work could explore further optimizations and extensions of the method, such as adaptive sparsity levels or dynamic parameter selection during fine-tuning.
* Conduct ablation studies to better understand the impact of each component of the proposed method. For instance, isolate the effects of sparsity, quantization, and ZO optimization to assess their individual contributions to the overall performance.
* Include more comprehensive comparisons with state-of-the-art fine-tuning methods, including both memory-efficient techniques and traditional methods. This would provide a clearer benchmark for evaluating the proposed method's relative performance.

---

### Official Review · Reviewer_HBxv · 2024-06-14
**Zeroth-Order Fine-Tuning of LLMs with Extreme Sparsity**

**Confidence:** 3

**Summary:**

The paper combines quantization and extreme sparsity with zeroth-order optimization to reduce memory consumption during fine-tuning of large language models (LLMs). This method fine-tunes a very small subset of LLM parameters using only the forward pass, allowing the majority of untuned LLM parameters to remain in a quantized state. By integrating these techniques, the proposed method enables fine-tuning models on devices with limited memory. The study uses a sensitive sparse parameters mask to identify an extremely small subset of the model's parameters for fine-tuning. The study found that fine-tuning just 0.1% of the LLM's sensitive parameters using ZO optimization outperforms full ZO fine-tuning while saving on wall-clock time.

**Strengths:**

- The study's motivation is clearly defined, emphasizing the need for memory-efficient fine-tuning of large language models (LLMs) on edge devices.
- The paper introduces a novel approach by combining zeroth-order (ZO) optimization with extreme sparsity and quantization. This unique combination reduces memory usage and computational costs, making it possible to fine-tune LLMs on devices with limited resources.
- The paper provides enough empirical evidence on how different sparse matrices as well as the percentage of LLM parameters to optimize, affect the ZO finetuning results.
- The results in the paper are supported by extensive experiments, making it a solid work and impactful contribution to the research community.
- The method was validated on the edge device. By enabling LLM fine-tuning on-edge devices, the proposed method has significant practical implications. It opens up new possibilities for deploying powerful LLMs in resource-constrained environments, expanding their usability and accessibility.

**Weaknesses:**

- The method requires identifying the sensitive parameters for each task. It is important to understand how much memory and time is required for this step.
- Figures 5 and 6: Adding the result or discussion on the result for other tasks (e.g. SST-2, CB, BoolQ, and WSC) and models (e.g. Mistral-7B and OPT-6.7B) will provide a more comprehensive result.

**Suggestions:**

- The text will need proper spacing at some lines to allow easy reading. e.g. Line 144 right column, Line 179.

---

### Meta-Review · Area_Chair_VDkC · 2024-06-14

**Recommendation:** Accept (Poster)
**Confidence:** 4

**Metareview:**

The reviewers are positive about the paper and its merits. I agree with their assessment and recommend acceptance.
I would also suggest and encourage the following:
1. It would be very helpful to explain the GPU memory demand for ZO fine-tuning in detail. (As reviewers suggested, to improve memory management in future work)
2. For 4-bit quantization on edge devices, since some of them support INT8, it would be worth exploring how much the proposed methods could benefit from INT8 in future work.

The author should clarify the meaning of "with Float16 datatype" and "16 bit". As (for throughput) official Flash Attention 2 does not support fp16, and (for fine-tuning) not all (edge) devices support bf16.

---

### Decision · Program_Chairs · 2024-06-17

**Decision:**

Accept (Poster)

**Comment:**

We thank the authors for their time and contribution to WANT and we are pleased to share that after the reviewing process the paper has been accepted. Congratulations! We encourage the authors to consider reviewers' feedback for the improvement of the camera-ready version. We hope to see you in person at the workshop and brainstorm on efficient training research together!